# Ordering of trajectories reveals hierarchical finite-time coherent sets in Lagrangian particle data: detecting Agulhas rings in the South Atlantic Ocean

David Wichmann[1,2], Christian Kehl[1], Henk A. Dijkstra[1,2], and Erik van Sebille[1,2]

[1]Institute for Marine and Atmospheric Research Utrecht, Utrecht University
[2]Centre for Complex Systems Studies, Utrecht University
**Correspondence:** David Wichmann (d.wichmann@uu.nl)

**Abstract.** The detection of finite-time coherent particle sets in Lagrangian trajectory data using data clustering techniques is an active research field at the moment. Yet, the clustering methods mostly employed so far have been based on graph partitioning, which assigns each trajectory to a cluster, i.e. there is no concept of noisy, incoherent trajectories. This is problematic for applications in the ocean, where many small coherent eddies are present in a large, mostly noisy fluid flow. Here, for the first time in this context, we use the density-based clustering algorithm OPTICS (Ankerst et al., 1999) to detect finite-time coherent particle sets in Lagrangian trajectory data. Different from partition-based clustering methods, derived clustering results contain a concept of noise, such that not every trajectory needs to be part of a cluster. OPTICS also has a major advantage compared to the previously used DBSCAN method, as it can detect clusters of varying density. The resulting clusters have an intrinsically hierarchical structure, which allows one to detect coherent trajectory sets at different spatial scales at once. We apply OPTICS directly to Lagrangian trajectory data in the Bickley jet model flow and successfully detect the expected vortices and the jet. The resulting clustering separates the vortices and the jet from background noise, with an imprint of the hierarchical clustering structure of coherent, small-scale vortices in a coherent, large-scale, background flow. We then apply our method to a set of virtual trajectories released in the eastern South Atlantic Ocean in an eddying ocean model and successfully detect Agulhas rings. We illustrate the difference between our approach and partition-based k-Means clustering using a 2-dimensional embedding of the trajectories derived from classical multidimensional scaling. We also show how OPTICS can be applied to the spectral embedding of a trajectory-based network to overcome the problems of k-Means spectral clustering in detecting Agulhas rings.

## 1 Introduction

Understanding the transport of tracers in the ocean is an important topic in oceanography. Despite large-scale transport features of the mean flow, on smaller scales, mesoscale eddies and jets play an important role for tracer transport (Van Sebille et al.,

2020). Such eddies can capture large amounts of a tracer, and, while transported in a background flow, redistribute them in the ocean. Eddies have been shown to play an important role for the accumulation of plastic (Brach et al., 2018) and the transport of heat and salt (Dong et al., 2014). To quantify the effects of eddies on tracer transport in the ocean, it is necessary to develop methods that are able to detect and track them. Many methods exist to detect such *finite-time coherent sets* of fluid parcels based on different mathematical or heuristic principles (Hadjighasem et al., 2017). The term 'finite-time coherent set' is based on the work of Froyland et al. (2010), and is in our context defined as a set of particles that stay, in a sense to be made more specific, close to each other along their entire trajectories. Here, for the first time in this context, we make use of the density-based clustering algorithm OPTICS (Ankerst et al., 1999) to detect finite-time coherent sets in Lagrangian trajectory data.

The detection of coherent Lagrangian vortices using abstract embeddings of Lagrangian trajectories together with data clustering techniques has received significant attention in the recent literature (Froyland and Padberg-Gehle, 2015; Hadjighasem et al., 2016; Padberg-Gehle and Schneide, 2017; Banisch and Koltai, 2017; Schneide et al., 2018; Froyland and Junge, 2018; Froyland et al., 2019). Using embedded trajectories for the detection of finite-time coherent sets is interesting as it allows one to use sparse trajectory data, and it can in principle be applied to ocean drifter trajectories, as demonstrated by Froyland and Padberg-Gehle (2015) and Banisch and Koltai (2017) for the detection of the five ocean basins. Yet, most of these methods cluster trajectory data with graph partitioning, which does not incorporate the difference between coherent, clustered trajectories and noisy trajectories that should not belong to any cluster. Graph partitioning has been shown to work in situations where the finite-time coherent sets are not too small compared to the fluid domain (Froyland and Padberg-Gehle, 2015; Hadjighasem et al., 2016; Padberg-Gehle and Schneide, 2017; Banisch and Koltai, 2017; Froyland and Junge, 2018). For applications to Lagrangian trajectory datasets on basin-scale ocean domains, where multiple small-scale coherent sets (eddies) coexist with noisy trajectories in the background, graph partitioning is however likely to fail. Similar observations were made by Froyland et al. (2019) for the partition-based clustering approaches based on transfer and dynamic Laplace operators (Froyland and Junge, 2018). Although some attempts have been made to accommodate such concepts in hard partitioning, e.g. by incorporating one additional cluster corresponding to noise (Hadjighasem et al., 2016), this approach is likely to fail for large ocean domains, as discussed by Froyland et al. (2019) and shown in section 4 of this paper. Froyland et al. (2019) have developed a special form of trajectory embedding based on sparse eigenbasis decomposition given the eigenvectors of transfer operators and dynamic Laplacians. By superposing different sparse eigenvectors, they successfully separate coherent vortices from unclustered background noise.

Motivated by the results Froyland et al. (2019) obtained by developing a new form of trajectory embedding, we here explore the potential of another clustering algorithm to overcome the inherent problems of partition-based clustering. We use the density-based clustering method OPTICS (Ordering Points To Identify the Clustering Structure) developed by Ankerst et al. (1999) to detect finite-time coherent sets in large ocean domains, using a very simple choice of embedding (cf. section 3.2.1). Density-based clustering aims to detect groups of data points that are close to each other, i.e. regions with high data *density*. Our data points correspond to entire trajectories, and groups of trajectories staying close to each other over a certain time interval correspond to such regions of high point density. Different from partition-based methods such as k-Means or fuzzy-c-means, OPTICS does not require to fix the number of clusters beforehand. Further, density-based clustering has an intrinsic notion of

a noisy data point: a point does not belong to any cluster (i.e. a finite-time coherent set) if it is not part of a dense region. A more detailed comparison of the method presented here to existing related methods can be found in section 3.4.

Another desirable property of the OPTICS algorithm is its ability to capture coherence hierarchies. In the ocean, coherent sets of trajectories naturally come with a notion of such a hierarchy. For example, the surface flow in the North Atlantic Ocean can be seen as approximately coherent (Froyland et al., 2014), while mesoscale eddies and jets are also finite-time coherent sets of trajectories at smaller scales *within* the North Atlantic Ocean. Froyland et al. (2019) show how their leading eigenvectors resolve coherent sets at large scales, while small-scale results can be obtained with a sparse eigenbasis approximation of a set of eigenvectors. Similarly, clustering results obtained from OPTICS are typically hierarchical. The main result of OPTICS, the reachability plot, provides this hierarchical information in a simple 1-dimensional graph.

In section 4, we first show how OPTICS detects finite-time coherent sets at different scales for the Bickley jet model flow (also discussed e.g. by Hadjighasem et al. (2017)), successfully detecting the six coherent vortices and the jet as the steepest valleys in the reachability plot. The general structure of the reachability plot also reveals the large-scale finite-time coherent sets, i.e. the northern and southern parts of the model flow, separated by the jet. We then apply our method to Lagrangian particle trajectories released in the eastern South Atlantic Ocean, where large rings detach from the Agulhas Current (e.g. Schouten et al. (2000)). We detect several Agulhas rings, and on the larger scale also separate the eastward and westward moving branches of the South Atlantic Subtropical Gyre. While the traditional approach to study Agulhas rings is based on sea surface height analysis (see e.g. Dencausse et al. (2010)), several methods based on virtual Lagrangian trajectories have been applied to Agulhas ring detection before (Haller and Beron-Vera, 2013; Beron-Vera et al., 2013; Froyland et al., 2015; Hadjighasem et al., 2016; Tarshish et al., 2018). Our method is different from these approaches in that it is directly applicable to a trajectory dataset, i.e. without much pre-processing of the data. As the OPTICS algorithm is readily available in the sklearn package of SciPy, the detection of finite-time coherent sets can be done without much effort and with only a few lines of code. A further difference is the mentioned intrinsic notion of coherence hierarchy, which allows for simultaneous analysis of trajectory data at different scales. While we mainly focus on the direct embedding of trajectories in an abstract high-dimensional Euclidean space, we also show in appendix C that OPTICS can be used to overcome the limits of k-Means clustering in the context of spectral clustering of the trajectory-based network of Padberg-Gehle and Schneide (2017).

## 2 Trajectory datasets

### 2.1 Quasi-periodically perturbed Bickley jet

We apply our method to a model system that has been used frequently in studies to detect finite-time coherent sets (Hadjighasem et al., 2017; Padberg-Gehle and Schneide, 2017; Hadjighasem et al., 2016; Banisch and Koltai, 2017; Froyland and Junge, 2018). The velocity field of the quasi-periodically perturbed Bickley jet (Bickley, 1937; del Castillo-Negrete and Morrison,

1993) is defined by a stream function $\psi(x,y,t)$, i.e. $\dot{x} = -\frac{\partial \psi}{\partial y}$ and $\dot{y} = \frac{\partial \psi}{\partial x}$, with $\psi(x,y,t) = \psi_0(y) + \psi_1(x,y,t)$ consisting of a stationary eastward background flow

$$\psi_0(y) = -UL\tanh(y/L), \tag{1}$$

and a time-dependent perturbation

$$\psi_1(x,y,t) = UL\operatorname{sech}^2(y/L)\operatorname{Re}\left[\sum_{n=1}^{3} f_n(t)\exp(ik_n x)\right], \tag{2}$$

where $\operatorname{Re}(z)$ denotes the real part of the complex number $z$. We use the same parameter values as Hadjighasem et al. (2017), with $U = 62.66$ m/s the characteristic velocity of the zonal background flow, and $L = 1770$ km. The parameters in eq. (2) are given by $k_n = 2n/r_0$, $f_n(t) = \epsilon_n \exp(-ik_n c_n t)$ with $\epsilon_1 = 0.075$, $\epsilon_2 = 0.4$, $\epsilon_3 = 0.3$, $c_1 = 0.1446U$, $c_2 = 0.205U$, $c_3 =$
$0.461U$. The domain of interest is $\Omega = [0, \pi r_0] \times [-3000\text{ km}, 3000\text{ km}]$, where $r_0 = 6371$ km is the radius of the Earth, and the left and right edges of $\Omega$ are identified, i.e. the flow is periodic in x-direction with period $\pi r_0$. Similar to Banisch and Koltai (2017), we seed the domain with an initial number of 12,000 particles on a uniform $200 \times 60$ grid. For this choice, the initial particle spacing is slightly above 100 km in both directions. We compute the trajectories for 40 days with a time step of one second using the SciPy integrate package. We output the trajectories every day, i.e. we have $T = 41$ data points in time for each
trajectory.

## 2.2   Agulhas rings in the South Atlantic

To test the OPTICS algorithm with a more realistic ocean flow, we simulate surface particle trajectories in a strongly eddying ocean model. Surface velocities are derived from a NEMO ORCA-N006 run (Madec, 2008), which has a horizontal resolution of $1/12°$ and velocity output for every five days. The model is forced by reanalysis and observed data of wind, heat and fresh
water fluxes (Dussin et al., 2016), i.e. the currents do not only contain the geostrophic component, as is the case in altimetry-derived currents (Beron-Vera et al., 2013; Froyland et al., 2019). For the advection of virtual particles, we use version 1.11 of the open source Parcels framework (Lange and van Sebille, 2017), see oceanparcels.org. The 2-dimensional surface current velocity is interpolated in space and time with the C-grid interpolation scheme of Delandmeter and van Sebille (2019), using a 4th order Runge-Kutta method with a time step of 10 minutes. We initially distribute particles uniformly in the ocean on
the vertices of a $0.2° \times 0.2°$ grid in the domain $[30°W, 20°E] \times [40°S, 20°S]$, which corresponds to a total number of 23,821 particles. At 30°S, a spacing of $0.2°$ corresponds to roughly 20 km. The particles start at January 5, 2000 and are advected for two years. We output the trajectories with a time interval of five days. We only use the first 100 days as data to detect the finite-time coherent sets, i.e. we have $T = 21$ data points for each trajectory, but also look at later times to see how long the rings need to disperse. We provide the used trajectory data for the Agulhas flow as NumPy file on Zenodo (Wichmann, 2020b).

## 3 Methods

### 3.1 Detecting coherent structures in Lagrangian trajectory data

For $N$ trajectories of dimension $D$ and length $T$, the trajectory information can be stored in a *data matrix* $X \in \mathbb{R}^{N \times DT}$, where each row results from a particle trajectory by concatenating the different spatial dimensions. The analysis of trajectory data to detect finite-time coherent sets of trajectories (Froyland and Padberg-Gehle, 2015; Banisch and Koltai, 2017; Hadjighasem et al., 2016; Padberg-Gehle and Schneide, 2017; Schneide et al., 2018; Froyland and Junge, 2018; Wichmann et al., 2020) can be split into two essential steps:

Step 1 **Embedding** of the trajectories in an abstract (metric) space, i.e. $X \to \bar{X} \in \mathbb{R}^{N \times M}$, where $M \leq DT$. If one uses a dimensionality reduction method, $M < DT$.

Step 2 **Clustering** of the embedded data with a clustering algorithm.

The embedding is necessary to represent the trajectories as points in a metric space. Different options for embedding the trajectories exist, e.g. a direct embedding of the data points along the trajectories (Froyland and Padberg-Gehle, 2015), or embeddings based on the eigenvectors derived from networks that are defined by physically motivated trajectory similarities (Banisch and Koltai, 2017; Padberg-Gehle and Schneide, 2017; Banisch and Koltai, 2017; Froyland and Junge, 2018). Once an embedding of each trajectory as a point in a metric (typically Euclidean) space is established, one can apply a clustering algorithm. Roughly speaking, clustering algorithms try to identify groups of points that are close to each other as a cluster. Partition-based clustering methods divide the entire data into a (typically fixed) number of $K$ clusters, such that each data point belongs to a cluster. The most popular method in this category is the k-Means algorithm, which tries to find a given number of $K$ clusters such that the sum of pairwise squared distances of points within a cluster is minimized. Other clustering algorithms contain a concept of 'noisy' data, i.e. data points that do not belong to any cluster, or belong to a cluster only with a certain probability. Examples for the former case are DBSCAN (Ester et al., 1996), discussed by Schneide et al. (2018) in the fluid dynamics context, and the here presented OPTICS (Ankerst et al., 1999) algorithm. For the latter case, the most popular method is fuzzy-c-means clustering, as discussed by Froyland and Padberg-Gehle (2015) in the context of finite-time coherent sets.

Figure 1 shows a few possible options for trajectory embedding and clustering that have partially been explored before (see the footnotes in the figure for the combinations used in related studies). For a given trajectory dataset, one can in principle apply an arbitrary combination of embedding and clustering methods. Only a few of the different combinations have been explored so far, and many more options for embedding and clustering as those shown in fig. 1 exist. It is important to note that a good choice of embedding and clustering might well depend on the specific problem at hand, and there might be no combination that performs well for all possible situations.

Most of the studies that use clustering techniques to detect finite-time coherent sets have focused on developing new forms of trajectory embeddings. For example, Hadjighasem et al. (2016), Padberg-Gehle and Schneide (2017), Banisch and Koltai (2017) and Froyland and Junge (2018) all use different forms of spectral embeddings, together with k-Means clustering.

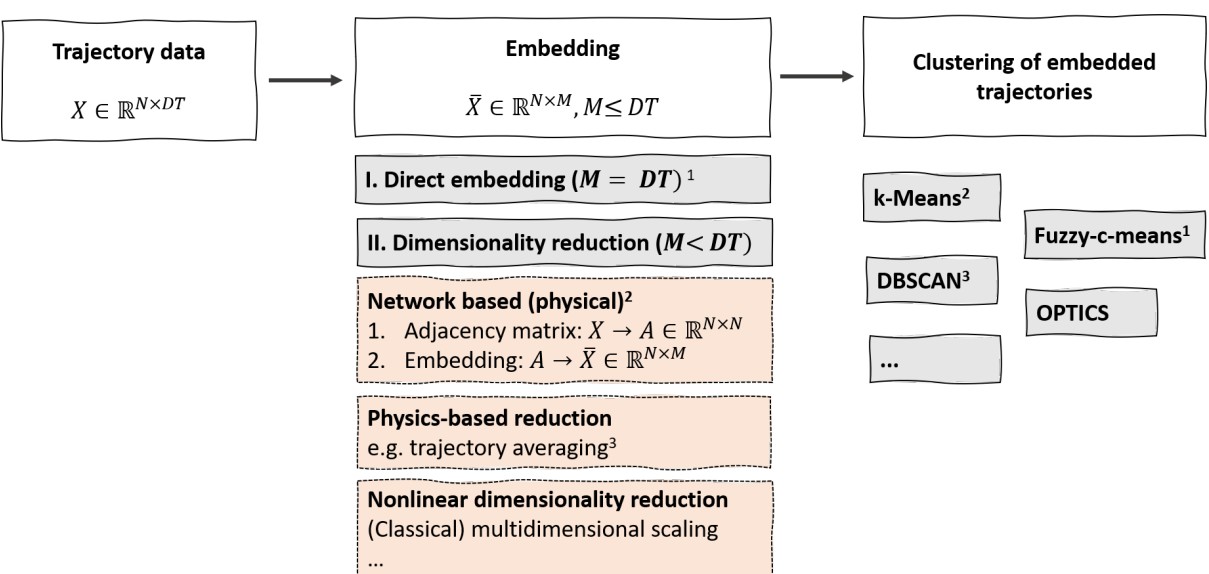

**Figure 1.** Different steps to detect coherent trajectories in Lagrangian data with trajectory clustering. The figure is non-exhaustive, and many more options for embedding and clustering exist. Footnotes: [1] Froyland and Padberg-Gehle (2015). [2] Hadjighasem et al. (2016), Padberg-Gehle and Schneide (2017) and Banisch and Koltai (2017) all define networks with spectral embedding and subsequent k-Means clustering. Froyland et al. (2019) define spectral embeddings defined on dynamic Laplacian and transfer operators. [3] Schneide et al. (2018).

Froyland et al. (2019) have developed a powerful form of embedding, based on a sparse eigenbasis approximation. Here, we focus on the clustering step in fig. 1, and propose the OPTICS clustering algorithm in the fluid dynamics context. We test the algorithm for three different kinds of embeddings:

E1  A direct embedding of the trajectory data in a high dimensional Euclidean space, i.e. $M = DT$ (cf. section 3.2.1).

E2  A reduction of the trajectory data to a 2-dimensional embedding space using classical multidimensional scaling (MDS, cf. section 3.2.2). This is mainly to visualize the difference to partition-based k-Means clustering.

E3  A spectral embedding of the network proposed by Padberg-Gehle and Schneide (2017).

In the following sections, we explain in detail the embeddings E1 and E2 and the OPTICS algorithm. We introduce the network embedding E3 together with the corresponding results in appendix C.

### 3.2 Trajectory embedding

#### 3.2.1 Direct embedding

The direct embedding of each trajectory in $\mathbb{R}^{DT}$ is the most straightforward embedding as it requires no further pre-processing of the trajectory data. For simplicity, assume we are given a set of $N$ trajectories in a 3-dimensional space, i.e. $(x_i(t), y_i(t), z_i(t))$

where $i = 1, \ldots, N$ and $t = t_1, \ldots, t_T$. We then simply define the embedding of trajectory $i$ in the abstract $3T$-dimensional space as

$$u_i = (x_i(t_0), x_i(t_1), \ldots, x_i(t_T), y_i(t_0), y_i(t_1), \ldots, y_i(t_T), z_i(t_0), z_i(t_1), \ldots, z_i(t_T)) \in \mathbb{R}^{3T}, \tag{3}$$

and impose an Euclidean metric in $\mathbb{R}^{3T}$ to measure distances between different embedded trajectories. The resulting embedded data matrix $\bar{X}$ is then simply given by the vertical concatenation of the different embedding vectors. This kind of embedding was also explored by Froyland and Padberg-Gehle (2015), together with a fuzzy-c-means clustering. Intuitively, if two trajectories $i$ and $j$ belong to the same finite-time coherent set, the corresponding particles follow very similar pathways, i.e. the Euclidean distance of the embedding vectors $d_{ij} = ||u_i - u_j||$ is expected to be small. On the other hand, a particle $i$ that belongs to a coherent set is expected to have a larger distance to a particle $j$ that is not part of the set. In other words, groups of particles that form a finite-time coherent set are *dense* in the embedding space. This motivates to use a density-based clustering algorithm to detect finite-time coherent sets.

To take into account the $\pi r_0$-periodicity in x-direction of the Bickley jet flow, we first put the individual 2-dimensional data points on the surface of a cylinder with radius $r_0/2$ in $\mathbb{R}^3$, and interpret the resulting trajectories in a 3-dimensional Euclidean space. The resulting data matrix is $\bar{X} \in \mathbb{R}^{N \times 3T}$, with $N = 12,000$ and $T = 41$. For the Agulhas particles, we put the single data points on the earth surface in a 3-dimensional Euclidean embedding space by the standard coordinate transformation of spherical to Euclidean coordinates. The resulting data matrix is thus $\bar{X} \in \mathbb{R}^{N \times 3T}$ with $N = 23,821$ and $T = 21$.

### 3.2.2 Dimensionality reduction with classical multidimensional scaling

To get an intuition for what the OPTICS algorithm does, and the differences to k-Means, we wish to visualize the data structure in the plane. For this, it is necessary to reduce the embedding dimension of each trajectory from $3T$ to two in a way that the density structure, and hence the individual Euclidean distances between embedded trajectories $d_{ij} = ||u_i - u_j||$, cf. eq. (3), are preserved. We do so by a common method of nonlinear dimensionality reduction, called classical multidimensional scaling (MDS), see e.g. chapter 10.3 of Fouss et al. (2016). Classical MDS tries to find an embedding of the high-dimensional data points in a low dimensional space such that the pairwise distances are approximately preserved. Similar to a principal component analysis, classical MDS makes use of the eigenvectors corresponding to the largest eigenvalues of a kernel matrix, which is in this case defined by

$$B = -\frac{1}{2} H \Delta^2 H, \tag{4}$$

where $\Delta^2 \in \mathbb{R}^{N \times N}$ is a matrix containing all squared distances between the points, $\Delta_{ij}^2 = ||u_i - u_j||^2$, and $H$ is the centring matrix with $H_{ij} = \delta_{ij} - 1/N$, where $\delta_{ij}$ denotes the Kronecker delta. The matrix $B$ in eq. (4) is called the centred inner product matrix. If $\tilde{B}$ is the matrix of inner products of the embedded data points, i.e. $\tilde{B}_{ij} = u_i \cdot u_j$ with Euclidean scalar product, then $B$ can be obtained by removing the mean of all rows and columns of $\tilde{B}$, cf. chapter 10.3 of Fouss et al. (2016). An embedding of

the data points using the eigenvectors corresponding to the leading non-negative eigenvalues of $B$ in eq. (4) ensures to capture the main variance of the (squared) distance structure, similar to a principal component analysis.

We compute $\Delta^2$ with the Euclidean embedding described in section 3.2.1 and restrict ourselves to the first two dimensions to visualize the data structure in the plane, i.e. the embedding is defined by

$$u_i = (w_{0,i}, w_{1,i}), \ i = 1, \ldots, N, \tag{5}$$

where $Kw_j = \lambda_j w_j$, and $\lambda_0 \geq \lambda_1 \geq \lambda_k$ for all $k = 2, \ldots N-1$. This choice of embedding ensures to capture the main variance of the data points, and we therefore also expect to capture the main structure in terms of data density. For large particle sets however, computing the spectrum of $H$ in eq. (4) is computationally not feasible, as the matrix $B$ is dense and computing the spectrum scales with $O(N^3)$. We apply classical MDS to the 12,000 particles of the Bickley jet model flow, and a random selection of the equal number of particles for the Agulhas flow. In our context, the method is most useful for visualization purposes, as it provides a good 2-dimensional approximation of the point distances, i.e. also the density structure of the embedded trajectories.

## 3.3 Clustering with OPTICS

The detection of dense accumulations of points that are separated from each other by non-dense regions (noise) is the main goal of density-based clustering. We use the OPTICS (Ordering Points To Identify the Clustering Structure) algorithm by Ankerst et al. (1999) to detect these regions. The OPTICS algorithm can be seen as an extension of DBSCAN (Ester et al., 1996). As we have no prior information on the density structure of the embedded nodes, we set the 'generating distance' of OPTICS to infinity and our presentation here is limited to this case. The general OPTICS algorithm with finite generating distance is computationally more efficient and slightly more complicated, and we refer to Ankerst et al. (1999) for more details.

For $\delta \in \mathbb{R}$, the $\delta$-neighbourhood of a point $p \in \mathbb{R}^M$ is defined as the $M$-dimensional ball of radius $\delta$ around $p$. Define $M_\delta(p)$ as the number of points that is in the $\delta$-neighbourhood of $p$, including $p$ itself. OPTICS requires one parameter, an integer $s_{min}$ (called MinPts by Ankerst et al. (1999)), that defines the *core-distance* of a point $p$ as

$$c(p) = \{\min(\delta) \mid M_\delta(p) \geq s_{min}\}. \tag{6}$$

The core distance is simply the minimum radius of a ball around $p$, such that the ball contains $s_{min}$ points. Note that the generating distance that we set to infinity is a maximum cut off distance for the computation of the core distance in eq. (6), beyond which the core distance is not defined. As we do not have an intuition for a good value of such a cut off, we remove it by setting it to infinity.

The ordering of the points is based on the *reachability distance* of a point $p$ w.r.t. another point $q$, defined as

$$r(p|q) = \max(c(q), ||p - q||), \tag{7}$$

where $||p - q||$ in our case denotes the Euclidean distance between $p$ and $q$. The ordering of points is then constructed with the following scheme:

Step 1 Pick a point $p_1$. This is the first point in the order, and is arbitrary.

Step 2 Compute the core-distance $c(p_1)$ of $p_1$.

Step 3 Define an ordered seed list containing all other points, $p_l$, $l = 2, \ldots, N$. For each point $p_l$, define the reachability value

$r(p_l)$ as the reachability distance (eq. (7)) w.r.t. $p_1$, $r(p_l) = r(p_l|p_1)$. Order the list in ascending order of the $r(p_l)$.

Step 4 Pick the first point on the ordered seed list as $p_2$ and compute the core-distance $c(p_2)$. For all remaining points $p_l$, $l = 3, \ldots, N$, update the reachability value $r(p_l) \to \min(r(p_l), r(p_l|p_2))$.

Step 5 Update the ordered seed list according to the new reachability.

Step 6 Repeat steps 4-5 to obtain $p_3$. Continue until all points are processed.

Note that the ordering of points is achieved by constantly updating the ordered seed list, cf. step 3. In this way, the algorithm iterates through groups of dense points one after the other, and only continues with other points once a dense region has been fully explored. Note also that the entire algorithm depends on the choice of the parameter $s_{min}$. The value of $s_{min}$ should be chosen roughly as a minimum value of the expected cluster size. In the examples presented in this paper, we take values for $s_{min}$ that correspond to the estimated minimum size of the coherent sets.

The main result of the OPTICS algorithm is a *reachability plot*. This plot is the graph defined by $(i, r(p_i))$, where $p_0 = \infty$ by definition. The reachability plot is a powerful presentation of the global and local distribution of a set of points at once. The valleys in this plot correspond to dense regions, which we relate to finite-time coherent sets. We show examples of reachability plots in section 4. Given the reachability plot $(i, r(p_i))$, we use two common ways to derive a clustering result:

 1. **DBSCAN clustering**: Choose a cut-off parameter $\epsilon$ and define all points $p_i$ with $c(p_i) \leq \epsilon$ as core points. All points

that are not in the $\epsilon$-neighbourhood of a core point are defined as noise. This set of noisy data points is equivalent to all points $p_i$ that are not core points and have a reachability value $r(p_i)$ with $r(p_i) > \epsilon$. A cluster of size $L$ is then defined as a consecutive set (in the sense of the ordering) of non-noise points $(p_j, p_{j+1}, \ldots, p_{j+L-1})$, with adjacent points $p_{j-1}$ and $p_{j+L}$ being noise. This is similar to the clustering result of a DBSCAN run with equal values for $s_{min}$ and $\epsilon$. All possible realizations of DBSCAN clusters, with the same value for $s_{min}$, can therefore be derived from the reachability

values, core distances and the ordering determined by OPTICS. Up to boundary points, a DBSCAN clustering result can be obtained by drawing horizontal lines in the reachability plot, cf. section 4.

 2. $\xi$-**clustering**: While the DBSCAN clustering method looks for deep valleys in the reachability plot, this method looks for valleys with steep boundaries. In short, the larger a parameter $\xi$ with $0 < \xi < 1$, the steeper the boundary of a valley has to be to be classified as a cluster. In more detail, a $\xi$-cluster is defined as a consecutive set of points $(p_j, p_{j+1}, \ldots, p_{j+L-1})$

that has steep boundaries in the sense that for a parameter $\xi$, $0 < \xi < 1$:

(a) The start of the cluster $p_j$ is in a $\xi$-steep downward area. A $\xi$-steep downward area is a maximal set of consecutive points $(p_l, p_{l+1}, \ldots, p_{l+k})$, $k \in \{1, \ldots, N-l\}$ where: 1. $p_l$ and $p_{l+k}$ are $\xi$-steep downward points, i.e. $r(p_l) \leq (1-\xi)r(p_{l-1})$ and $r(p_{l+k}) \leq (1-\xi)r(p_{l+k-1})$, 2. $p_{l+i} \leq p_l$ for all $i = 1, \ldots, k$ and 3. not more than $s_{min}$ consecutive points in the set are no $\xi$-steep downward points.


(b) The end of the cluster $p_{j+L-1}$ is a $\xi$-steep upward area. The definitions are the reverse of the $\xi$-steep downward area, with the definition of a $\xi$-steep upward point as $r(p_j) \leq (1-\xi)r(p_{j+1})$.

(c) The cluster contains at least $s_{min}$ points, i.e. $L \geq s_{min}$.

(d) Every point in the inside of the cluster is at least a factor of $(1-\xi)$ smaller than the boundary points $p_j$ and $p_{j+L-1}$. All points that do not belong to a cluster are classified as noise.

We refer to Ankerst et al. (1999) for a more detailed discussion of the $\xi$-clustering method with illustrations for example data. Note that the full $\xi$-clustering method presented by Ankerst et al. (1999) does contain some more details related to the choice of the start and end points, which we did not mention here.

The OPTICS algorithm as well as functions to derive both clustering results from an OPTICS output are available in the SciPy sklearn package. Note that the implementation in sklearn allows for a minimum cluster size different from $s_{min}$ for the $\xi$-

clustering method (item 2 c above), but we will not make use of this additional freedom to reduce the number of parameters. Note that, different from k-Means, both clustering methods do not require an a priori determination of the number of clusters. For the $\xi$-clustering method, a larger $\xi$ requires steeper boundaries to form a cluster, i.e. will typically lead to a reduction of the number of resulting clusters. For DBSCAN clustering with very large $\epsilon$, one will detect one large global cluster. Making $\epsilon$ smaller leads then to consecutive splits of this cluster, forming (up to noise) a cluster hierarchy. We will demonstrate the

properties for both clustering methods in section 4 for different situations. In the following applications, we use an estimation of the minimum number of particles per finite-time coherent set for the parameter $s_{min}$.

Intuitively, the two clustering methods can be understood as follows. DBSCAN detects those groups of points that have a certain minimum density defined by the minimum reachability distance $\epsilon$. Clusters detected by DBSCAN are therefore defined by a global density criterion. This assumes no structural differences in the type of coherent sets in different regions of the fluid.

Different from that, the $\xi$-clustering method detects clusters by finding strong changes in the density of the data points, and not based on absolute densities. This has the advantage that clusters of different absolute density can be detected. Such a situation can arise if the distribution of particles is inhomogeneous over the fluid domain, or if the spatial extend of the fluid domain is very large such that the properties of finite-time coherent sets vary significantly. It is important to note that the main result of OPTICS is the reachability plot itself. The DBSCAN- and $\xi$-clustering methods should be seen as useful tools to identify the

most important features of that plot.

### 3.4   Comparison to related methods

Our method is closely related to existing methods to detect finite-time coherent sets with clustering techniques. Most notably, Froyland and Padberg-Gehle (2015) also use a direct embedding of individual trajectories similar to eq. (3), together with fuzzy-

c-means clustering. Hadjighasem et al. (2016), Banisch and Koltai (2017), Padberg-Gehle and Schneide (2017) and Froyland and Junge (2018) use spectral embeddings of graphs that are defined on some form of physical intuition or of dynamical operators, together with k-Means clustering. These studies show applications of their methods to example flows where the size of almost-coherent sets is not too small compared to the fluid domain. Such examples are the Bickley jet flow, which we also study in section 4.1, the five major ocean basins (Froyland and Padberg-Gehle, 2015; Banisch and Koltai, 2017), or few individual eddies in an ocean or atmospheric flow (Hadjighasem et al., 2016; Padberg-Gehle and Schneide, 2017; Froyland and Junge, 2018). In such situations, noisy background trajectories can be detected as individual clusters by the partitioning method, as discussed by Hadjighasem et al. (2016). For applications in large ocean domains, where the number of eddies is not known beforehand and where there are many more noisy trajectories than coherent trajectories, such an approach is likely to fail, see also the discussion by Froyland et al. (2019). OPTICS does not require to fix the number of clusters beforehand, and also contains an intrinsic concept of noisy trajectories that do not belong to any cluster, making OPTICS suitable for challenging flows in large domains.

As mentioned, OPTICS also contains an intrinsic notion of cluster hierarchy, i.e. coherent sets that are themselves part of coherent sets at larger scales. Ma and Bollt (2013) studied hierarchical coherent sets in the transfer operator framework of Froyland et al. (2010), in the spirit of the hierarchical clustering method proposed by Shi and Malik (2000). Their approach is also partition-based, i.e. there is no concept of noisy trajectories. In addition, at each stage of the hierarchy, a fixed cut-off has to be chosen based on minimizing an objective function (Ma and Bollt, 2013). Different from that approach, the main result of OPTICS, the reachability plot, contains such hierarchical information in a smooth and intrinsic manner.

As described in section 3.3, clustering results of the DBSCAN algorithm (Ester et al., 1996) can be derived from the reachability plot of OPTICS. DBSCAN has been used in the context of coherent sets before by Schneide et al. (2018), although not to identify specific clusters, but to distinguish noisy from clustered trajectories. The potential of density-based clustering for applications in the ocean and its comparison to other existing clustering methods for flow examples such as the Bickley jet (cf. section 2.1) has not been explored so far. Different from OPTICS, DBSCAN detects clusters with a certain fixed minimum density, although clusters with varying densities might be present in a dataset (Ankerst et al., 1999). More specifically, the value for the cut-off parameter $\epsilon$, cf. section 3.3, has to be set beforehand. Choosing a good value for the density parameter in DBSCAN is challenging if there is no underlying physical intuition for the density structure. As described in section 3.3, OPTICS allows one to derive any DBSCAN clustering result, with the same value for the parameter $s_{min}$, after computing the reachability plot, i.e. after one can get first insights into the clustering structure of the dataset to make an appropriate choice for $\epsilon$. Furthermore, it also allows one to use the $\xi$-clustering method instead of DBSCAN (cf. section 3.3).

A more recent and powerful technique to detect finite-time coherent sets in sparse trajectory data was presented by Froyland et al. (2019), based on dynamic Laplacian and transfer operators (Froyland and Junge, 2018). Froyland et al. (2019) apply their method to a trajectory dataset in the Western Boundary Current region in the North Atlantic Ocean, and successfully detect many eddies by superposing individual eigenvectors. The methods presented there are based on a form of spectral embedding, derived from discretized dynamical operators. Based on this embedding, clustering results have also been derived with k-Means by Froyland and Junge (2018) and with individual thresholding by Froyland et al. (2019). Froyland et al. (2019) also show how

the low-order eigenvectors correspond to large-scale coherent features, while the individual eddies are derived by a sparse eigenbasis approximation of a number of eigenvectors. The latter approach is essentially a transformation of the embedding to represent the most reliable features, such that a superposition of the eigenvectors alone yields the information about the location and size of finite-time coherent sets (without a clustering step). This is essentially an optimized form of embedding, i.e. the second step in fig. 1. Our aim here is to focus on the third step in fig. 1, i.e. to demonstrate the potential of the density-based clustering algorithm OPTICS, together with a very simple embedding of eq. (3).

A downside of our method compared to other approaches is the rather ad-hoc choice of embedding, cf. eq. (3). Different from many other methods, most notably the ones of Banisch and Koltai (2017), Froyland and Junge (2018) and Froyland et al. (2019), this type of embedding is not derived from a meaningful dynamical operator. It could be fruitful to explore a combination of these more meaningful embeddings together with OPTICS as a clustering algorithm in future research.

## 4  Results

### 4.1  Bickley jet flow

We start with the direct embedding of the Bickley jet flow trajectories, cf. section 2. The data matrix has dimension $X \in \mathbb{R}^{12,000 \times 123}$. We apply the OPTICS algorithm to the resulting points together with DBSCAN clustering, choosing $s_{min} = 80$ as a minimum size of the finite-time coherent sets. In the following, all axis units are in multiples of $1000$ km. Figure 2 shows the reachability plot, together with the DBSCAN clustering result of three different choices of $\epsilon$. The six vortices and the jet are clearly visible as the major valleys in the reachability plot. The hierachical structure of the DBSCAN clustering with decreasing $\epsilon$ is visible in the figures from top (large-scale coherence) to bottom (small-scale coherence). Note that for the DBSCAN clustering results, boundary points of the clusters can be above the hozitonal line at $y = \epsilon$. This is because of the definition of the DBSCAN clustering in section 3.3.

To illustrate the difference between OPTICS and k-Means, we use the embedded trajectories and apply classical MDS to obtain a 2-dimensional embedding. As described in section 3.2.2, this assures to capture the major variance along the embedding axes. The spectrum of $B$ in eq. (4) is shown in fig. A1 in the appendix, with two clearly dominant eigenvalues. The fact that there are two very dominant eigenvalues assures that the illustration of the data in the plane captures the major variance of the data points. Figure 3a shows the corresponding embedding of the trajectories in the 2-dimensional Euclidean space. The star-shaped distribution of data points reflect the strong symmetries of the underlying idealized Bickley jet flow. Such symmetry is not expected to be present for more realistic flows. Figures 3b and 3c show the cluster labels for OPTICS with DBSCAN clustering at $\epsilon = 10^6$ km, and for a k-Means clustering with $K = 8$ clusters, respectively. $K = 8$ corresponds to the six vortices, the jet, and one noise cluster as suggested by Hadjighasem et al. (2016).

The corresponding clustering results in real space are shown in figs. 4 and 5 for OPTICS and k-Means, respectively. The jet and the six vortices are clearly recognizable as dense accumulations of points in the 2-dimensional space of fig. 3b, see fig. 4 for the corresponding colours. The clustering result with k-Means in fig. 5 shows that the clusters corresponding to the vortices are much less focussed. In addition, each of the eight clusters in fig. 3c contains some of the noisy points of fig. 3b, which

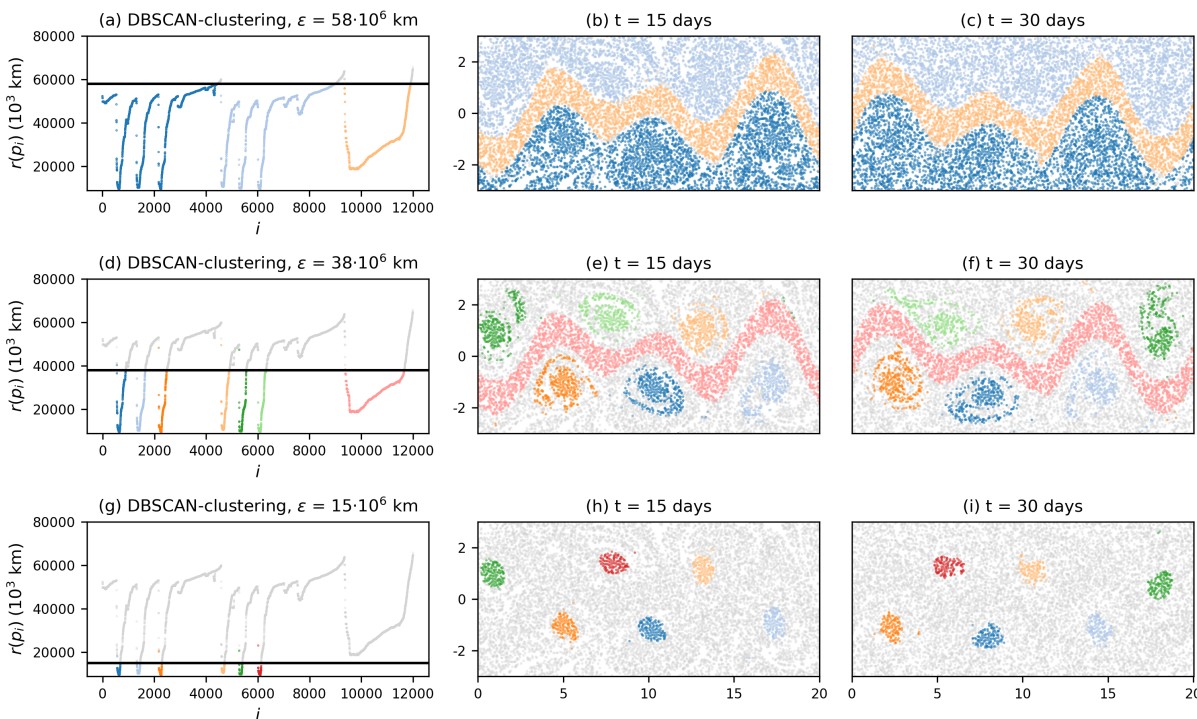

**Figure 2.** Result of the OPTICS algorithm applied to the direct embedding of the trajectories. (a), (d) and (g) show the reachability plot with different DBSCAN clustering results, indicated by the black horizontal line. The corresponding clustering results of each choice of DBSCAN parameter $\epsilon$ is shown on the right of the reachability plots for different times. Grey particles correspond to noise. Axis units in the centre and right column are in 1000 km.

shows that using one additional cluster for noise does not work in this situation. It is interesting to note that capturing the noisy data points of fig. 3b by an additional cluster in k-Means is geometrically impossible, simply because k-Means clusters are circular. Covering all noisy points without including the centre, i.e. the jet in fig. 3b, is not possible for k-Means.

It should be noted here that the poor performance of k-Means in figs. 3c and 5 is not representative for other methods that use k-Means. For example, the method of Banisch and Koltai (2017) captures the coherent structures in the Bickley jet rather well, including the jet in the middle. We emphasize again that we use classical MDS here mostly for visualization purposes, as the computation of the classical MDS embedding is difficult for large particle sets. In our case, a dense $12,000 \times 12,000$ symmetric matrix has to be diagonalized, which already takes a significant amount of computation time.

We finally also tested the performance of our algorithm with a random subset of 2,000 particles, using data for every five days instead of every day, cf. fig. A2 in the appendix. OPTICS still detects the six vortices and the jet, although the cluster boundaries are less clearly defined compared to fig. 2. Froyland and Junge (2018) detect the vortices and the jet by using data of 3,000 particles only at initial and final times ($t = 0$ and $t = 40$ days). Our method is not able to detect the expected finite-time

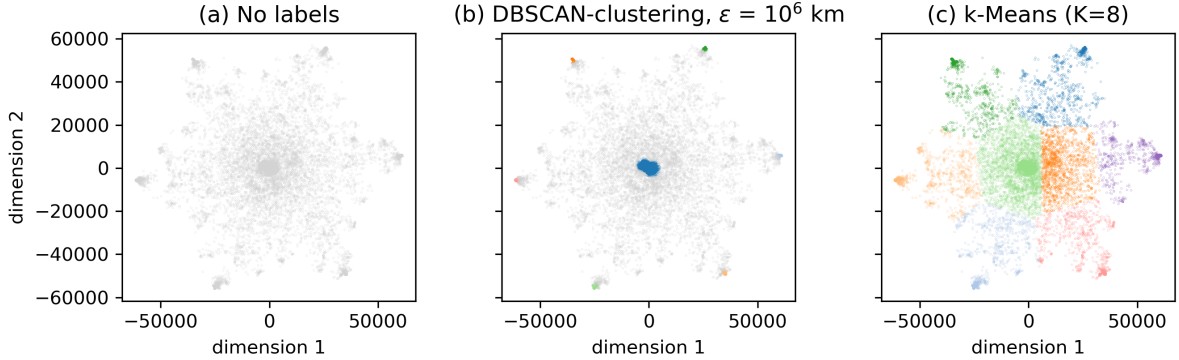

**Figure 3.** a: 2-dimensional embedding of the classical MDS method (cf. section 3.2.2) of the trajectories. b: with labels according to the DBSCAN result of fig. 4. The six vortices and the jet are clearly visible as dense regions. Grey particles correspond to noise. c: k-Means clustering result for K=8, see fig. 5 for the spatial clustering result of k-Means.

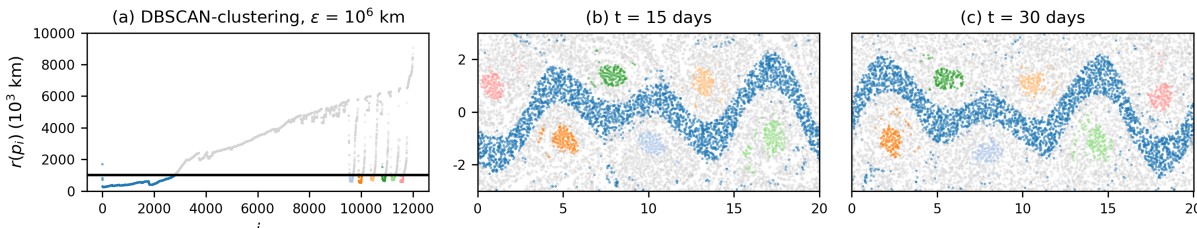

**Figure 4.** Result of DBSCAN clustering of the 2-dimensional embedding of the classical MDS method. a: reachability plot with black line representing the DBSCAN parameter $\epsilon$. b-c: corresponding clustering results at different times. Grey particles represent noise. Axis units are in 1000 km.

coherent sets with using only initial and final particle data. This is likely to be a result of the ad-hoc direct embedding, cf. eq.
(3), see the discussion at the end of section 3.4.

## 4.2   Agulhas rings

We next apply OPTICS to the Agulhas trajectories. As described in section 2, we have $\bar{X} \in \mathbb{R}^{N \times 63}$ with $N = 23,821$. We choose $s_{min} = 100$ in the following, which corresponds initially to a square cell of $2° \times 2°$, i.e. a reasonable minimum size of an Agulhas ring. Figure 6 shows the result of the direct embedding. The reachability plot in fig. 6a is much more jagged
than for the Bickley jet model flow (cf. fig. 2a). The narrow deep valleys and the wider valleys in the reachability plot indicate the presence of large- and small-scale coherence patterns. Figure 6a-c show the DBSCAN clustering result for a relatively large value of $\epsilon$. The main separation of fluid domains is between the red and the blue particles, with a few vortices at their boundary. These two water masses are the northern and southern parts of the subtropical gyre in the South Atlantic, the red

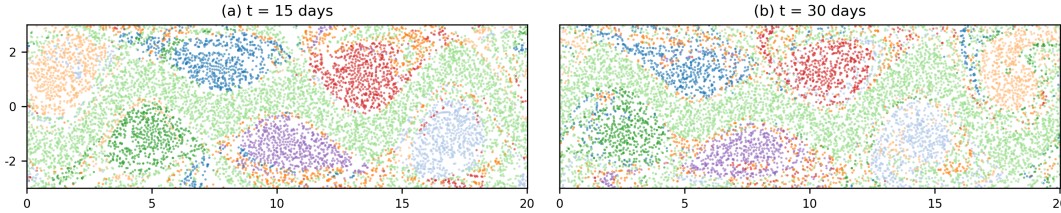

**Figure 5.** Result of $K = 8$ k-Means clustering of the 2-dimensional embedding from classical MDS, cf. fig. 4. Axis units are in 1000 km.

particles moving to the west, the blue particles to the east. The second and third rows of fig. 6 show other clustering results
for the DBSCAN- and the $\xi$-clustering method, respectively. The valleys in fig. 6g with steepest boundaries as detected by
the $\xi$-clustering method mostly correspond to eddy-like structures, separated by background noise. Note that not all clusters
in the figure correspond to eddies. For example, the blue cluster in figs. 6g-i stays approximately coherent over the considered
time interval, although it is certainly not an Agulhas ring. An animation of the detected finite-time coherent sets for the full
two years of trajectory data based on the $\xi$-clustering method as in the last row of fig. 6 can be found on Zenodo (Wichmann,
2020a), showing that many of the sets stay coherent for significantly longer times than the first 100 days.

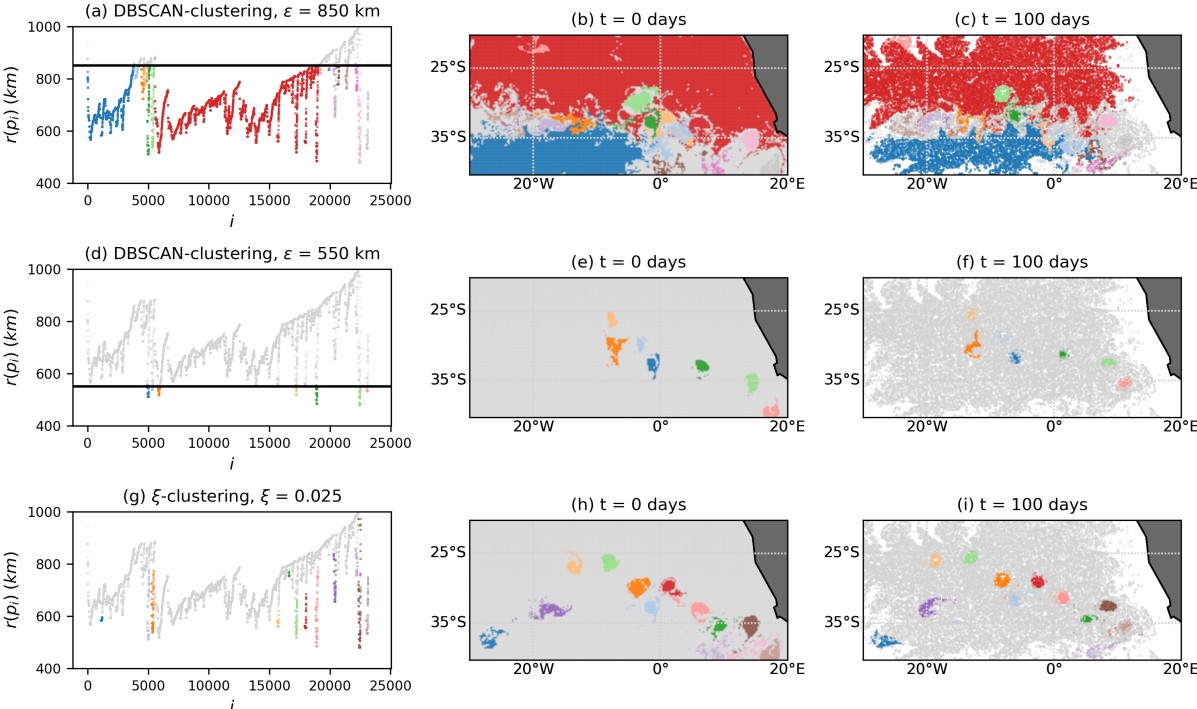

**Figure 6.** Result of the OPTICS algorithm applied to the direct embedding of the trajectories, with different clustering methods. Grey
particles correspond to noise.

Figure 6 shows that for this situation, the $\xi$-clustering method detects more Agulhas rings than DBSCAN. While the clustering results shown in the figure all depends on the parameter values for $\xi$ and $\epsilon$, it is visible in the reachability plot of fig. 6g that the definition of some eddies includes the entire boundary of the valleys, i.e. up to very high reachability values. At the same time, the detection of the large-scale clusters as in 6a-c is not possible with the $\xi$-clustering method. These findings are in fact expected, cf. the discussion of the two clustering methods at the end of section 3.3. DBSCAN is best to detect global density structures, i.e. when the reachability values of all points are compared to the same cut-off $\epsilon$. Regions that are dense locally but not necessarily globally are better detected with the $\xi$-clustering method. Despite these differences between the two clustering methods, we again emphasize that the main result of OPTICS is the reachability plot itself. Fig. 7 shows a colour map at initial time of the reachability values. We clearly see Agulhas rings as the dark regions corresponding to lowest values of reachability. The regions of large reachability correspond to trajectories that are relatively noisy compared to all the other trajectories.

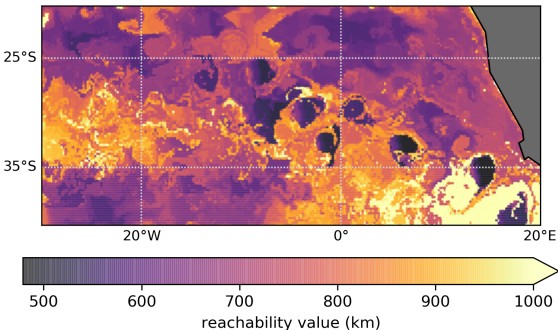

**Figure 7.** Reachability values at initial time, resulting from the OPTICS algorithm applied to the direct embedding of the trajectories. The regions with lowest values clearly correspond to Agulhas rings. The colour bar is cut off at a reachability of 1000 km to show the relevant structure of variations.

In order to illustrate again the difference between OPTICS and k-Means for this example, we choose 12,000 random trajectories and again embed the trajectories in a 2-dimensional space with classical MDS (cf. section 3.2.2). The reduction of the particle set is necessary to simplify the eigendecomposition of the matrix $B$ in eq. (4), and we therefore choose $s_{min} = 30$. The corresponding spectrum of $B$ is shown in fig. B1 in the appendix, showing that there are again two dominant eigenvectors, i.e. visualizing the netwok in the plane captures the main variance of the data. Figure 8 shows the embedded trajectories together with OPTICS / DBSCAN clustering (fig. 8b) and k-Means (fig. 8c) for K=40. Figs. 9 and 10 show the corresponding clustering results in the fluid domain. It is visible that k-Means does not detect a single vortex, but splits the fluid domain into regions of approximately similar size. OPTICS detects multiple Agulhas rings by finding the deepest valleys in the reachability plot.

It is interesting to note that the use of classical MDS in fig. 9 has lead to the detection of many of the vortices of fig. 6d-f with DBSCAN instead of the $\xi$-clustering method. The transformation to the reduced 2D space has hence lead to a simplification of the reachability plot, which now represents the major variations in the distances of the embedded trajectories. At the same time, the large-scale structure of 6a is not visible any more in fig. 9. This indicates that exploring more dimensionality reduction

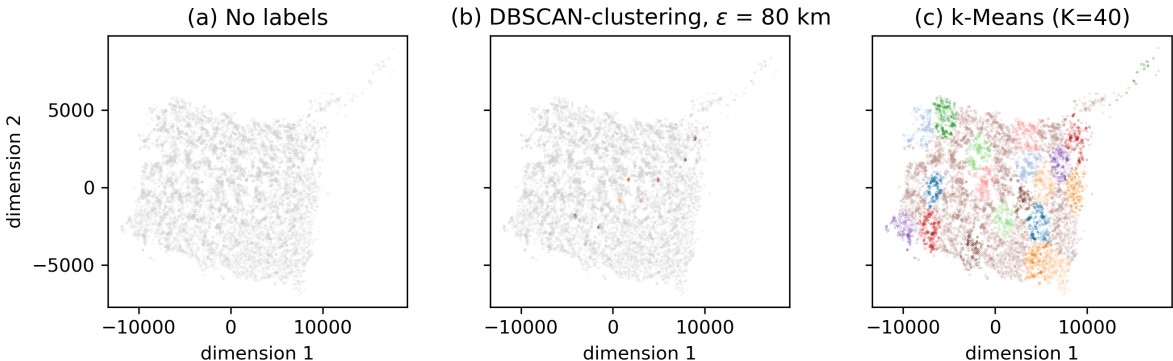

**Figure 8.** Embedding of the Agulhas trajectories in the 2-dimensional space defined by the leading eigenvectors of the MDS Kernel matrix $B$. a: no labels. b: clustering labels of OPTICS / DBSCAN, see fig. 9 for the corresponding plot in the Agulhas region. Grey particles represent noise. c: k-Means with $K = 40$, see fig. 10 for the corresponding plot in the Agulhas domain.

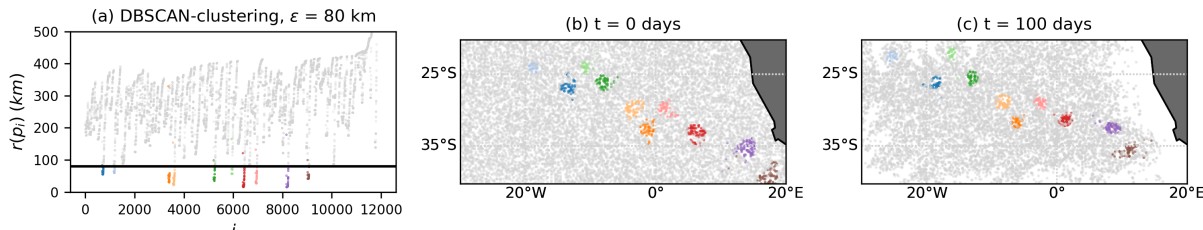

**Figure 9.** Result of OPTICS applied to the 2-dimensional embedding of 12,000 randomly selected particles with the classical MDS method, cf. fig. 8b, and $s_{min} = 30$. The corresponding spectrum is shown in fig. B1 in the appendix, showing that there are two dominant eigenvectors. Grey particles are classified as noise.

techniques could be useful for future research, in particular those that are computationally more efficient than classical MDS.

Spectral embeddings derived from networks together with partition-based clustering have a similar problem as the one illustrated in figs. 8c and 10 (Froyland et al., 2019). Similar to the case discussed here, OPTICS can be used to overcome the problems of k-Means. We show this in appendix C for the network proposed by Padberg-Gehle and Schneide (2017) for the Agulhas region, together with a brief introduction of the network and how to construct spectral embeddings. In summary, k-Means again fails to detect any of the vortices, while OPTICS detects many of the coherent vortices in the spectrally embedded

network. Yet, other flow features are also present that result from the physical motivation of the network definition, see the results in appendix C.

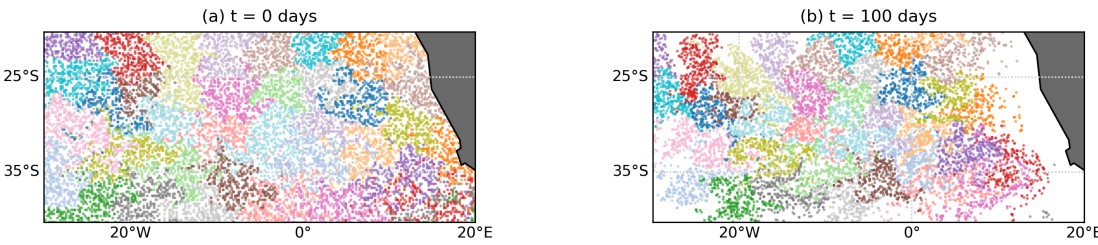

**Figure 10.** Result of the k-Means clustering with $K = 40$ applied to the 2-dimensional embedding with classical MDS, cf. fig. 8c.

## 5 Conclusions

The abstract embedding of particle trajectories in a metric space with subsequent clustering is a promising field of research for the detection of finite-time coherent sets in oceanography. Yet, most of the existing methods have been based on graph partitioning, which has no concept of noisy, unclustered trajectories. This is a problem for applications in the ocean, where many eddies are transported in a noisy background flow on large domains. This study is motivated by the success of Froyland et al. (2019) in overcoming the problem of graph partitioning by a sophisticated form of trajectory embedding. Here, we show how the density-based clustering algorithm OPTICS (Ankerst et al., 1999) can be used instead of graph partitioning, in order to detect small-scale eddies in large ocean domains. Different from partition-based clustering methods such as k-Means, OPTICS does not require to fix the number of clusters beforehand. Clusters are detected by identifying dense accumulations of points, i.e. groups of trajectories that are close to each other in embedding space. Coherent groups of particle trajectories can be identified as valleys in the reachability plot computed by the OPTICS algorithm. This plot also has a natural interpretation in terms of cluster hierarchies, i.e. finite-time coherent sets that are by themselves part of a larger scale finite-time coherent set. Such hierarchies are present in the surface ocean flow, where the subtropical basins are approximately coherent and at the same time contain other finite-time coherent structures such as eddies and jets.

We apply OPTICS to Lagrangian particle trajectories directly, in the spirit of Froyland and Padberg-Gehle (2015). OPTICS successfully detects the expected coherent structures in the Bickley jet model flow, separating the six vortices and the jet from background noise. We also apply OPTICS to simulated trajectories in the eastern South Atlantic and successfully identify Agulhas rings, separated by noise. We visualize the difference between OPTICS and k-Means with a 2-dimensional embedding of the trajectories based on classical multidimensional scaling. We also show how OPTICS can be applied to the spectral embedding of the particle-based network proposed by Padberg-Gehle and Schneide (2017), providing a necessary amendment to their method to detect coherent vortices in a large ocean domain, i.e. when k-Means fails. Our method is very simple to implement in Python, as OPTICS is available in the SciPy sklearn package. While we here present the results of OPTICS with three different kinds of embeddings, it is likely that OPTICS also works for other trajectory embeddings, such as the spectral embeddings of Banisch and Koltai (2017) or Froyland and Junge (2018). Using such dynamically motivated embeddings instead of the ad-hoc direct embedding presented here could be a promising direction for future research.

Extending our method to datasets with more trajectories can be made more efficient by choosing a finite generating distance for

OPTICS (Ankerst et al., 1999). While this is better from a computational point of view, it requires some knowledge or intuition about the spatial distribution of the embedded trajectories. A major challenge for the method proposed here is the embedding dimension. For long trajectories, it is necessary to reduce the dimensionality of the trajectories before applying OPTICS. A complication here is the desired property of an embedding to preserve both local and global distances in order to make full use of the hierarchical properties of OPTICS. This means, for example, that the popular method of a locally linear embedding (Roweis and Saul, 2000) is not suitable, unless only the small-scale (densest finite-time coherent sets) are to be detected. Using classical multidimensional scaling (MDS), as we did here to visualize the clustering results, in principle preserves local and global distances, although our results indicate that the large-scale coherence structure in the Agulhas flow is less pronounced for the classical MDS embedding compared to the full embedding of trajectories. In any case, classical MDS is not an option for very large datasets, as it requires the diagonalization of a dense symmetric square matrix of size equal to the particle number. Spectral embeddings of derived networks such as the ones of Hadjighasem et al. (2016), Padberg-Gehle and Schneide (2017) and Banisch and Koltai (2017) are useful to achieve lower-dimensional embeddings, but they come with the introduction of additional parameters for the network construction and heuristics to truncate the embedding dimension. Further research into other non-linear dimensionality reduction techniques that have not been explored in the context of finite-time coherent sets can lead to more efficient and robust methods.

*Code and data availability.* All code is available at https://github.com/OceanParcels/coherent_vortices_OPTICS, including the code to generate the Bickley jet trajectories. The data for the virtual particles in the South Atlantic is available on Zenodo (Wichmann, 2020b). Details on the Parcels simulation for the virtual trajectories in the ocean can be found at the GitHub repository of our previous paper, https://github.com/OceanParcels/near_surface_microplastic. The data from the NEMO ORCA-006 run are available at http://opendap4gws.jasmin.ac.uk/thredds/nemo/root/catalog.html

**Appendix A: Additional figures for the Bickley jet flow**

**Appendix B: Additional figures for the Agulhas flow**

**Appendix C: Detecting Agulhas rings with a particle-based network**

To demonstrate that OPTICS can also be applied to the spectral embedding of a particle-based network, we use the network proposed by Padberg-Gehle and Schneide (2017). If we have a set of particle trajectories $x_i(t)$, where $i = 1, \ldots, N$, $t = t_1, t_2, \ldots, t_T$ with $N$ the number of particles and $T$ the number of time steps, the network $A \in \mathbb{R}^{N \times N}$ is defined as:

$$A_{ij} = \begin{cases} 1, & \text{if } \exists t \in \{t_1, t_2, \ldots, t_T\} \ s.t. \ ||x_i(t) - x_j(t)|| < d, \\ 0, & \text{otherwise.} \end{cases} \tag{C1}$$

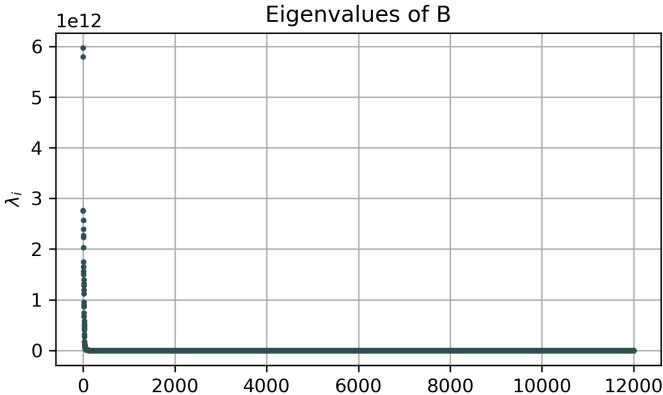

**Figure A1.** Spectrum of the classical MDS kernel matrix $B$ for the Bickley jet flow. It is visible that there are two dominant eigenvalues. We choose the vectors corresponding to these first two eigenvalues as embedding vectors in section 4.1.

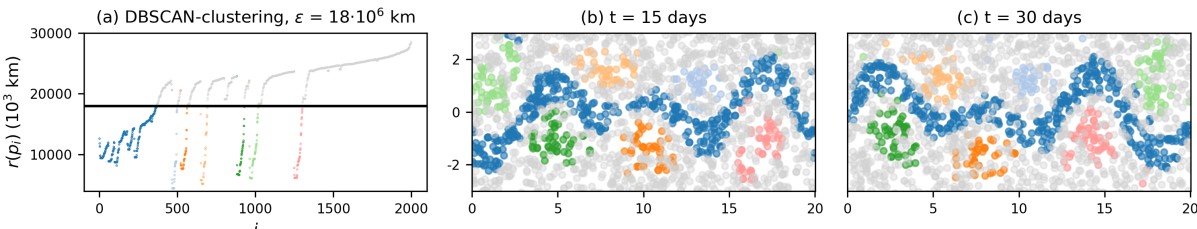

**Figure A2.** Result of the OPTICS algorithm for a random subset of 2,000 particles in the Bickley jet flow, with particle data every 5 days instead of every day. To account for the smaller number of particles, we set $s_{min} = 15$ for this case. The six vortices and the jet are still clearly visible.

Here, $||.||$ denotes the Euclidean norm and $d > 0$ is a fixed pre-determined cut-off parameter, see Padberg-Gehle and Schneide (2017) for a discussion on the choice of $d$ (called $\epsilon$ in Padberg-Gehle and Schneide (2017)). Similar to Padberg-Gehle and Schneide (2017), we embed the nodes in a lower dimensional space $\mathbb{R}^K$ by means of the eigenvectors of its random walk Laplacian, (see e.g. Von Luxburg (2007))

$$L_r = D^{-1}A, \tag{C2}$$

where $D$ is a diagonal matrix with $D_{ii} = \sum_j A_{ij}$. The embedding of node $i$ is defined by

$$y_i = (v_{1,i}, v_{1,i}, \ldots, v_{K,i}) \in \mathbb{R}^K, \tag{C3}$$

where $v_i, i = 0, \ldots, N-1$ are the right eigenvectors corresponding to the largest eigenvalues $\lambda_i$ of $L_r$. The eigenvalues are assumed to be ordered in descending order, i.e. $1 = \lambda_0 > \lambda_1 \geq \ldots, \geq \lambda_N$. The classical simultaneous K-way normalized cut

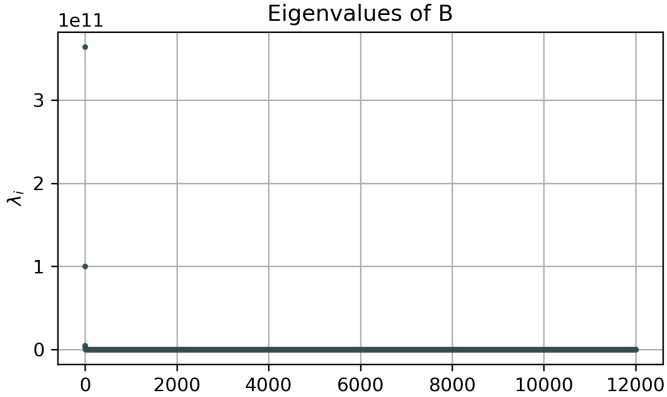

**Figure B1.** Spectrum of the classical MDS kernel matrix $B$ for the Agulhas flow, where we first constrain the particle data to 12,000 randomly selected trajectories. There are again two dominant eigenvalues, for which we choose the corresponding vectors for the embedding in section 4.2.

proceeds with applying the k-Means algorithm to the embedding defined in eq. (C3) to detect $K$ clusters (Von Luxburg, 2007),
resulting in an approximate solution to the normalized cut problem (Shi and Malik, 2000).

Figure C1 shows the spectrum of the resulting random walk Laplacian with $d = 200$ km. No obvious spectral gap is visible that would suggest a truncation of the embedding space. Figure C2 shows the clustering result if we apply a k-Means algorithm as suggested by Padberg-Gehle and Schneide (2017) to detect $K = 40$ clusters. It is visible that the partition-based k-Means clustering method does not detect any individual Agulhas rings, but partitions the state space into regions of approximately
equal size.

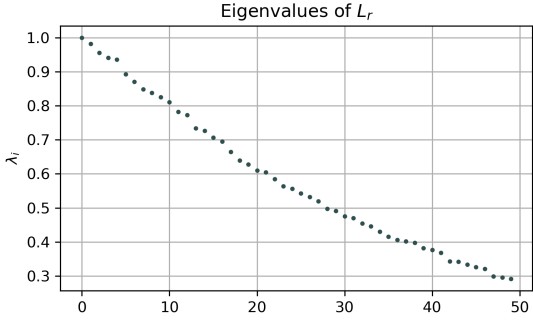

**Figure C1.** Spectrum of the random walk Laplacian, cf. eq. (C2) of the network proposed by Padberg-Gehle and Schneide (2017) applied to the Agulhas trajectory data. No clear gap exists that suggest a truncation of the embedding.

Applying OPTICS instead of k-Means with a subsequent $\xi$-clustering detects some of the Agulhas rings, see fig. C3, where we choose $s_{min} = 100$ as in section 4.2. Note that also other structures than typical circular eddies are detected. While this depends on the clustering parameter $\xi$ (or $\epsilon$ for DBSCAN), this is also a consequence of the *physically motivated* network

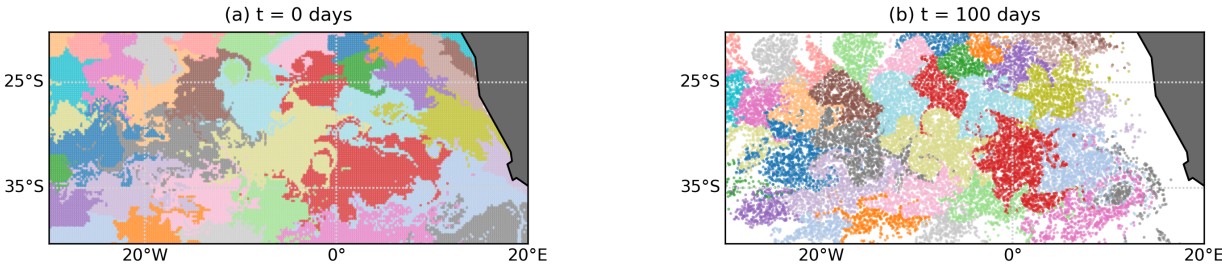

**Figure C2.** Result of k-Means clustering applied to the 40 leading eigenvectors of the random walk Laplacian, cf. eq. (C2), looking for 40 clusters. No individual vortices are detected.

defined by eq. (C3), where particles are connected equally if they are close to each other at least once in time. This is different from the direct embedding, where we require particles to stay close to each other along the entire trajectory.

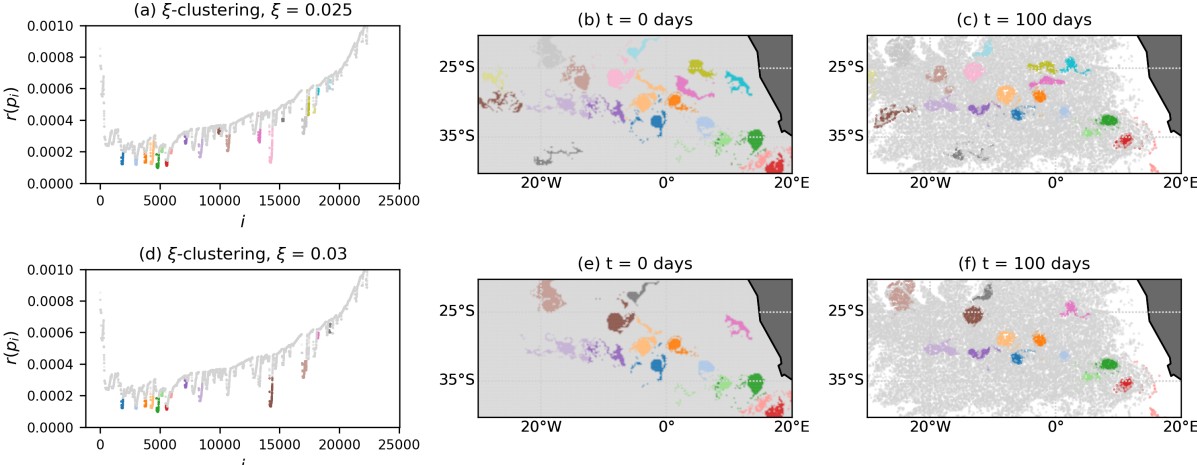

**Figure C3.** Result of OPTICS applied to the $K = 40$ spectral embedding of the network defined in eq. (C1) with $d = 200$ km and $s_{min} = 100$. Grey particles are classified as noise.

*Author contributions.* DW performed the analysis, with support from CK, EvS and HD. DW wrote the manuscript and all authors jointly edited and revised it.

*Competing interests.* The authors declare no competing interests

*Acknowledgements.* David Wichmann, Christian Kehl and Erik van Sebille are supported through funding from the European Research Council (ERC) under the European Union Horizon 2020 research and innovation programme (grant agreement No 715386). This work was partially carried out on the Dutch national e-infrastructure with the support of SURF Cooperative (project no. 16371). We thank Andrew Coward for providing the ORCA-N006 simulation data.

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
