# Peer review of "Ordering of trajectories reveals hierarchical finite-time coherent sets in Lagrangian particle data: detecting Agulhas rings in the South Atlantic Ocean"

_Nonlinear Processes in Geophysics, 2020_

## Referee Comment (RC1) · Anonymous Referee #1 · 25 Aug 2020

General comments

Overall, I find this paper to contribute potentially useful technical refinements of clustering methods for Lagrangian trajectories. The paper offers a modest but meaningful contribution, and is wisely concise in its presentation. The reachability plot does seem to be a useful conceptual tool, and the ability to admit incoherent or 'noise' regions is a nice refinement.

I find there to be room for improvement on a few presentational issues. (i). There

seems to be an assumption of familiarity with other clustering methods. The paper would be more accessible, and therefore useful, if the authors took just slightly more time in defining new terms and in providing the intuitive content of mathematical concepts. (ii) I find it a little strange that some figures are presented in the appendix, but discussed only in the main text. Some of these make good illustrations of the performance of the method with respect to others, e.g. D1&D2. I feel this tends to negatively impact the narrative. If figures are discussed in the main text, I would present them there also. (iii) The paper has a highly technical focus throughout. More framing of the import of this problem at the start and end would have been appreciated. (iv) For a short paper, the abstract is perhaps disproportionately long.

There is one point raised in the paper that I felt required more elaboration. A selling point the authors bring up for this method is that it can in principle be applied to real-world trajectory data, see line 86 and also line 306. This is true but incomplete. Real-world Lagrangian instruments are sufficiently sparse that it is rare to find more than one in the same eddy at the same time. Thus, the application presented herein—finding eddies is idealized configurations—is not really relevant for how one would apply this method to real-world trajectories. The data density used here is orders of magnitude greater than for real-world instruments. Since the authors bring this up as an advantage of the method, a more fair and nuanced discussion of its potential and limitations with respect to real-world data is called for.

I would say, rather, that the method seems more suitable in application to model data or virtual trajectories from altimetry, where it benefits from a simplicity with respect to some other proposed methods.

Specific comments

Line 99. Do you not want to cite Bickley?

W.G. Bickley D.Sc. (1937) LXXIII. The plane jet , The London, Edinburgh, and Dublin Philosophical Magazine and Journal of Science, 23:156, 727-731, DOI:

10.1080/14786443708561847

My understanding is that the term "Bickley jet" itself is used to refer to a steady solution with a sechˆ2 u-velocity, see e.g. Swaters (1999). The authors' Eq. (2) is an added perturbation. As read, it sounds like the whole thing is the Bickley jet.

Section 3.2.2. I didn't really understand this section, or what B is encoding in Eq. (4). A more intuitive description would be helpful. When you say, "pairwise distances are approximately preserved", this is with respect to what? Also, why are two dimensions chosen?

Line 193. The intuitive meaning of the 'generating distances' that are not being used here should be mentioned.

Line 196. The definition of the epsilon neighborhood appears incomplete. Is it not the M-dimensional sphere of radius epsilon? Otherwise, what is the epsilon?

Line 200. It would be very helpful to write out in words the meaning of Eq. (6). My understanding is that $c(p)$ is minimum distance epsilon such that the number of points in an epsilon neighborhood is greater than a specified number.

Line 213. I did not immediately understand how it arises that there are valleys in the reachability if you have sorted iteratively on the reachability. You might explain that this happens as you encounter groups of points that are all near to each other, thus replacing earlier high values of reachability with lower values.

Line 216. The phrasing here made me wonder if this was a second, different epsilon. It would be clearer to say that you choose a value for the parameter epsilon. Also, it appears this is conditional on a choice of s_min which should then be emphasized.

Line 228. What are the permissible values of k in condition (a)?

Figure 2, what are the units of the y-axis in the left column of plots?

Figs 2 and 3, some of the colored dots lie above the epsilon threshold.

Figure 4. I really don't understand the two dimensions of these plots, nor the star-shaped patterns, could you explain these more?

Data locations at Zenodo should be cited, not only the papers referring to them.

Minor typographic comments

Throughout the paper, the authors consistently omit the subject ahead of an infinitive, e.g. "which allows to detect". I believe this is grammatically incorrect (in US usage anyway). "allows one to detect" or "allowing the detection of" sound better.

l 42 and 90. "sparse" should probably be used instead of "scarce". The former means thinly distributed while the latter means hard to come by.

l 128. NumPy and Zenodo are the standard capitalizations

l 141. "method" should be "methods"

l 156. straightforward

l 191. "and as will become clear"

l 217. "is equal to" should be "set of points is equivalent to".

l 243. "a priory" should be "a priori"

l 279. "large- and small-scale"

l 354. GitHub

l 359. There is a title of an appendix with no appendix.

l 360 & 361. "particle-based"

l 383. "ot"

l 389. There should be a period at the end of this sentence

Figure C1, "three" eigenvalues should be "two", correct?

[Figure]

---

## Referee Comment (RC2) · Anonymous Referee #2 · 23 Sep 2020

The authors apply a well-known clustering method OPTICS to Lagrangian trajectory data to extract finite-time coherent sets. Two of their aims are (i) to develop a hierarchy of coherent sets and (ii) to not fully partition the entire domain into coherent sets.

Novelty: ========

There are already several clustering methods in the literature for finding finite-time coherent sets, including a density-based clustering DBSCAN by Schneide-etal'18, which is a special case of the OPTICS approach in the manuscript. The idea of a hierarchy of

finite-time coherent sets has been considered by Ma/Bollt'13. The paper Fr/Sa/Ro'19 develops a robust method to classify only those sets are that coherent, not fully partitioning the domain. In Fr/Sa/Ro'19, coherent sets at different spatial scales are also considered, similar to a hierarchy. Fr/Sa/Ro'19 also considers the Bickley jet and ocean eddies, with ocean eddies listed as a motivation in Fr/Sa/Ro'19 for developing a non-partitioning approach. Not limited to the work above, I would say there is some "upselling" of the novelty in the manuscript, and that prior work is occasionally omitted, mischaracterized, or overly criticized.

A positive aspect is that the (standard) "DBSCAN" and "\xi" clustering outputs of the OPTICS clustering could provide potentially useful hierarchical information, and to my knowledge this is a new way of analyzing the dynamics. Unfortunately, this is not explored much, and the authors do not provide an intuitive explanation of what the "DBSCAN" and "\xi" clustering algorithms are actually doing in their dynamical context. It would be beneficial for the authors to link the algorithms more with the dynamical inputs (trajectories) and the dynamical problem being solved. As this is the main contribution of the paper, I think this needs to be expanded much more. The reasons behind the choices of which clustering algorithm is applied to the different datasets should also be explained.

Performance: ============

The (uncited) paper Froyland/Junge'18 develops a finite-element approximation of the dynamic Laplacian, which is a very accurate and robust method of finite-time coherent set extraction for low-dimensional systems of the type treated in the Wichmann manuscript. In Froyland/Junge'18 there are no free parameters, the method is unaffected by the density of the data points, and estimates are produced on the whole domain.

A comparison can be made for the Bickley example in the Wichmann manuscript because the setup is identical. Wichmann et al uses a 200x60 grid of points and particle

positions at times t=0,1,2,3,...,39,40. Froyland/Junge'18 studied the same Bickley flow as in Wichmann, except that Froyland/Junge'18 used a coarser 100x30 grid of points and only particle positions at time 0 and time 40. Figure 15 in Froyland/Junge'18 shows much clearer images with fewer trajectory inputs. Thus, I think there is not a strong case for the approach in the manuscript being a better performer.

The idea to not fully partition the domain has already been treated in Fr/Sa/Ro'19. Regarding the ocean eddy example in the manuscript, Fr/Sa/Ro'19 also applied the method of Froyland/Junge'18 to ocean flow and successfully extracted a greater number of eddies than Wichmann at a higher quality. On the other hand, Fr/Sa/Ro'19 used AVISO-derived trajectories rather than model output, so it could be that Wichmann is using a rougher velocity field. Wichmann also used lower trajectory density than Fr/Sa/Ro'19 by a factor of about 4; both of these items could make Wichmann's task more difficult, compared to Fr/Sa/Ro'19.

---

## Author Comment (AC1) · 19 Oct 2020

**Answer to reviewer 1**

**General answer:**
We thank the reviewer for the detailed comments on the paper. They have helped us to significantly improve on the readability and clarity in the revised version. We have implemented changes for every comment raised by the reviewer.

**Please note:**
The images in this file are excerpts of the revised version in latexdiff. Please apologize the formatting problems of latexdiff that cuts off references at line breaks. This is not the case in the revised version.
* * *
**Comment 1**
I find there to be room for improvement on a few presentational issues. (i). There seems to be an assumption of familiarity with other clustering methods. The paper would be more accessible, and therefore useful, if the authors took just slightly more time in defining new terms and in providing the intuitive content of mathematical concepts.

**Answer to comment 1**
Thank you for your comment. We have made the following changes in the revised version:

1. Additional paragraph in the methods section that briefly describes why embedding / clustering is necessary, and also explains in one sentence what k-Means does

> The embedding is necessary to represent the trajectories as points in a metric space. Different options for embedding the
> 160 trajectories exist, e.g. a direct embedding of the data points along the trajectories (Froyland and Padberg-Gehle, 2015), or
> embeddings based on the eigenvectors derived from networks that are defined by physically motivated trajectory similarities
> (Banisch and Koltai, 2017; Padberg-Gehle and Schneide, 2017; Banisch and Koltai, 2017; Froyland and Junge, 2018). Once an
> embedding of each trajectory as a point in a metric (typically Euclidean) space is established, one can apply a clustering
> algorithm. Roughly speaking, clustering algorithms try to identify groups of points that are close to each other as a cluster.
> 165 Partition-based clustering methods divide the entire data into a (typically fixed) number of $K$ clusters, such that each data
> point belongs to a cluster. The most popular method in this category is the k-Means algorithm, which tries to find a given
> number of $K$ clusters such that the sum of pairwise squared distances of points within a cluster is minimized. Other clustering
> algorithms contain a concept of 'noisy' data, i.e. data points that do not belong to any cluster, or belong to a cluster only with a
> certain probability. Examples for the former case are DBSCAN (Ester et al., 1996), discussed by Schneide et al. (2018) in the
> 170 fluid dynamics context, and the here presented OPTICS (Ankerst et al., 1999) algorithm. For the latter case, the most popular
> method is fuzzy-c-means clustering, as discussed by Froyland and Padberg-Gehle (2015) in the context of finite-time coherent
> sets.

2. Additional explanation in the methods section that describes why the embedding we choose is expected to create a detectable signal for OPTICS.

**3.2 Trajectory embedding**

**3.2.1 Direct embedding**

The direct embedding of each trajectory in $\mathbb{R}^{DT}$ is the most  straightforward embedding as it requires no further pre-processing of the trajectory data. For simplicity, assume we are given a set of $N$ trajectories in a 3-dimensional space, i.e.
$(x_i(t), y_i(t), z_i(t))$ where $i = 1, \ldots, N$ and $t = t_1, \ldots, t_T$. We then simply define the embedding of trajectory $i$ in the abstract $3T$-dimensional space as

$$u_i = (x_i(t_0), x_i(t_1), \ldots, x_i(t_T), y_i(t_0), y_i(t_1), \ldots, y_i(t_T), z_i(t_0), z_i(t_1), \ldots, z_i(t_T)) \in \mathbb{R}^{3T}, \qquad (3)$$

and impose an Euclidean metric in $\mathbb{R}^{3T}$ to measure distances between different embedded trajectories. The resulting embedded data matrix $\bar{X}$ is then simply given by the vertical concatenation of the different embedding vectors. This kind of embedding was also explored by Froyland and Padberg-Gehle (2015), together with a fuzzy-c-means clustering. Intuitively, if two trajectories $i$ and $j$ belong to the same finite-time coherent set, the corresponding particles follow very similar pathways, i.e. the Euclidean distance of the embedding vectors $d_{ij} = \|u_i - u_j\|$ is expected to be small. On the other hand, a particle $i$ that belongs to a coherent set is expected to have a larger distance to a particle $j$ that is not part of the set. In other words, groups of particles that form a finite-time coherent set are *dense* in the embedding space. This motivates to use a density-based clustering algorithm to detect finite-time coherent sets.

To take into account the $\pi r_0$-periodicity in x-direction of the Bickley jet flow, we first put the individual 2-dimensional data points on the surface of a cylinder with radius $r_0/2$ in $\mathbb{R}^3$, and interpret the resulting  trajectories in a 3-dimensional Euclidean space. The resulting data matrix is $\bar{X} \in \mathbb{R}^{N \times 3T}$, with $N = 12,000$ and $T = 41$. For the Agulhas particles, we put the single data points on the earth surface in a 3-dimensional Euclidean embedding space by the standard coordinate transformation of spherical to Euclidean coordinates. The resulting data matrix is thus $\bar{X} \in \mathbb{R}^{N \times 3T}$ with $N = 23,821$ and $T = 21$.
* * *
**Comment 2**

(ii) I find it a little strange that some figures are presented in the appendix, but discussed only in the main text. Some of these make good illustrations of the performance of the method with respect to others, e.g. D1&D2. I feel this tends to negatively impact the narrative. If figures are discussed in the main text, I would present them there also.

**Answer to comment 2**

Thank you for this comment. We agree with the reviewer and have now included the clustering results of the classical MDS method in the main text. In the revised version, we provide the results of OPTICS together with its comparison to k-Means for both of the model flows. We have decided to leave the discussion of the embedded network of Padberg-Gehle and Schneide (2017) together with the previous figures D1-D3 in the appendix. This is because the major focus of the paper is the OPTICS clustering on the direct embedding of the trajectories, as this removes the need of several parameters compared to Padberg-Gehle and Schneide (2017), such as the cut-off parameter *d*, and the embedding dimensions. A reader that is interested in the application of OPTICS to the spectral embedding of Padberg-Gehle and Schneide (2017) gets a full account on that topic in the appendix. We do not discuss these results in the main text, but only mention them quickly. The actual discussion is contained in appendix C of the revised version.

**Comment 3**

(iii) The paper has a highly technical focus throughout. More framing of the import of this problem at the start and end would have been appreciated.

**Answer to comment 3**

Thank you for this suggestion. We have now added more content on the problem itself, i.e. the detection of many small coherent structures in a large, noisy ocean domain.

**1. Introduction**

[revised manuscript text omitted]

**Comment 5**

There is one point raised in the paper that I felt required more elaboration. A selling point the authors bring up for this method is that it can in principle be applied to real-world trajectory data, see line 86 and also line 306. This is true but incomplete. Real-world Lagrangian instruments are sufficiently sparse that it is rare to find more than one in the same eddy at the same time. Thus, the application presented herein—finding eddies is idealized configurations—is not really relevant for how one would apply this method to real-world trajectories. The data density used here is orders of magnitude greater than for real-world instruments. Since the authors bring this up as an advantage of the method, a more fair and nuanced discussion of its potential and limitations with respect to real-world data is called for. I would say, rather, that the method seems more suitable in application to model data or virtual trajectories from altimetry, where it benefits from a simplicity with respect to some other proposed methods.

**Answer to comment 5**

The reviewer is correct that an application of our method to real drifters to detect eddies is not possible due to the limited coverage of drifter data. Note that two studies applied their methods to real drifters, as we mentioned in the introduction (Froyland and Padberg-Gehle (2015) and Banisch and Koltai (2017)), however to detect the five major ocean basins and not eddies. In the new version, we omit the reference to real ocean drifters at other places but the introduction, where we now explicitly mention the application to ocean basins (and not eddies).

1. Changes in introduction to clarify that trajectory-based clustering has been applied to real drifter data only in the context of detecting the ocean basins, not individual eddies.

The detection of coherent Lagrangian vortices using abstract embeddings of Lagrangian trajectories together with data clustering
40 techniques has received significant attention in the recent literature (Froyland and Padberg-Gehle, 2015; Hadjighasem et al., 2016; Padb
. Examples include the direct embedding of trajectories in a high dimensional Euclidean space (Froyland and Padberg-Gehle, 2015)
, or more abstract embeddings based on related networks constructed from particle trajectories (Hadjighasem et al., 2016; Padberg-Gehl
(Froyland and Padberg-Gehle, 2015; Hadjighasem et al., 2016; Padberg-Gehle and Schneide, 2017; Banisch and Koltai, 2017; Schne
. Using embedded trajectories for the detection of finite-time coherent sets is interesting as it allows to use searce one to use
45 sparse trajectory data, and it can in principle be applied to ocean drifter trajectories, as done demonstrated by Froyland and
Padberg-Gehle (2015) and Banisch and Koltai (2017) for the detection of the five ocean basins. Yet, the methods proposed so

2. End of the introduction

In section 4, we first show how OPTICS detects finite-time coherent sets at different scales for the Bickley jet model flow
95 (also discussed e.g. by Hadjighasem et al. (2017)), successfully detecting the six coherent vortices and the jet as the steepest
valleys in the reachability plot. The general structure of the reachability plot also reveals the large-scale finite-time coherent
sets, i.e. the northern and southern parts of the model flow, separated by the jet. We then apply our method to Lagrangian
particle trajectories released in the eastern South Atlantic Ocean, where large rings detach from the Agulhas Current (e.g.
Schouten et al. (2000)). We detect several Agulhas rings, and on the larger scale also separate the eastward and westward
100 moving branches of the South Atlantic Subtropical Gyre. While the traditional approach to study Agulhas rings is based
on sea surface height analysis (see e.g. Dencausse et al. (2010)), several methods based on virtual Lagrangian trajectories
have been applied to Agulhas ring detection before (Haller and Beron-Vera, 2013; Beron-Vera et al., 2013; Froyland et al.,
2015; Hadjighasem et al., 2016; Tarshish et al., 2018). Our method is different from these approaches in that it is directly
applicable to a trajectory data set dataset, i.e. without much pre-processing of the data. As the OPTICS algorithm is read-
105 ily available in the sklearn package of SciPy, the detection of finite-time coherent sets can be done without much effort and
with only a few lines of code. A further difference is the mentioned intrinsic notion of coherence hierarchy, which allows for
simultaneous analysis of trajectory data at different scales. Finally, trajectory-based approaches can in principle be applied
to searce trajectory data, i.e. to any Lagrangian particle simulation result without much care for the spatial coverage of the
initial conditions. While we mainly focus on the direct embedding of trajectories in an abstract high-dimensional Euclidean
110 space, we also show in section C in the appendix appendix C that OPTICS can be used to overcome the limits of k-Means
clustering in the context of spectral clustering of physically motivated trajectory-based networks, such as the works presented
by Hadjighasem et al. (2016), Padberg-Gehle and Schneide (2017) or Banisch and Koltai (2017) the trajectory-based network
of Padberg-Gehle and Schneide (2017).

2. First sentence in conclusion

The abstract embedding of particle trajectories in a metric space with subsequent clustering is a promising field of research
470 for the detection of finite-time coherent sets in oceanography, as it can be potentially applied to sparse sets of trajectories e.g.
from drifter release experiments. . Yet, most of the existing methods lack the ability to separate finite-time coherent structures
* * *
**Comment 6**
Line 99. Do you not want to cite Bickley? My understanding is that the term "Bickley jet" itself is used to refer to a steady solution with a sechˆ2 u-velocity, see e.g. Swaters (1999). The authors' Eq. (2) is an added perturbation. As read, it sounds like the whole thing is the Bickley jet.

**Answer to comment 6**
Thank you for this comment and the careful check of our references of the flow. You are indeed right that the Bickley Jet is a steady, sechˆ2 velocity profile. We have added the reference to Bickley now,

together with a reference to the paper of del Castillo-Negrete and Morrison (1993), where the perturbed form of the jet is motivated.
* * *
**Comment 7**

Section 3.2.2. I didn't really understand this section, or what B is encoding in Eq. (4). A more intuitive description would be helpful. When you say, "pairwise distances are approximately preserved", this is with respect to what? Also, why are two dimensions chosen?

**Answer to comment 7**

Thank you for this comment. In the new version, we elaborate more on the intuitive goal of classical MDS in this section. We choose two dimensions because we wish to visualize the data in the plane. We have made this more clear in the new version.

**3.2.2  Dimensionality reduction with classical multidimensional scaling**

To get an intuition for what the OPTICS algorithm does, and the differences to k-Means, we wish to visualize the data structure in the  plane. For this, it is necessary to reduce the embedding dimension of each trajectory from $3T$ to two in a way that

220 the density structure, and hence the individual Euclidean distances between embedded trajectories $d_{ij} = \|u_i - u_j\|$, cf. eq. (3), are preserved. We do so by a common method of nonlinear dimensionality reduction, called classical  multidimensional scaling (MDS), see e.g. chapter 10.3 of Fouss et al. (2016). Classical MDS tries to find an embedding of the high-dimensional data points in a low dimensional space such that the pairwise distances are approximately preserved.

 Similar to a principal component analysis, classical MDS makes use of the eigenvectors corresponding to the largest

225 eigenvalues of  a kernel matrix, which is in this case defined by

$$B = -\frac{1}{2} H \Delta^2 H,\tag{4}$$

where $\Delta^2 \in \mathbb{R}^{N \times N}$ is a matrix containing all squared distances between the points, $\Delta_{ij}^2 = \|u_i - u_j\|^2$, and $H$ is the centring matrix with $H_{ij} = \delta_{ij} - 1/N$, where $\delta_{ij}$ denotes the Kronecker delta. The matrix $B$ in eq. (4) is called the centred inner product matrix. If $\bar{B}$ is the matrix of inner products of the embedded data points, i.e. $\bar{B}_{ij} = u_i \cdot u_j$ with Euclidean scalar product, then

230 $B$ can be obtained by removing the mean of all rows and columns of $\bar{B}$, cf. chapter 10.3 of Fouss et al. (2016). An embedding of the data points using the eigenvectors corresponding to the leading non-negative eigenvalues of $B$ in eq. (4) ensures to capture the main variance of the (squared) distance structure, similar to a principal component analysis.

We compute $\Delta^2$ with the Euclidean  embedding described in section 3.2.1 and restrict ourselves to the first two dimensions to visualize the data structure in the plane, i.e. the embedding is defined by

235 $$u_i = (w_{0,i}, w_{1,i}), \ i = 1, \dots, N,\tag{5}$$

where $K w_j = \lambda_j w_j$, and $\lambda_0 \geq \lambda_1 \geq \lambda_k$ for all $k = 2, \dots N - 1$. This choice of embedding ensures to capture the main variance of the data points, and we therefore also expect to capture the main structure in terms of data density. For large particle sets however, computing the spectrum of $H$ in eq. (4) is computationally not feasible, as the matrix $B$ is  dense and computing the spectrum scales with $O(N^3)$. We apply classical MDS to the 12,000 particles of the Bickley jet model flow,

240 and a random selection of the equal number of particles for the Agulhas flow. In our context, the method is most useful for visualization purposes, as it provides a good 2-dimensional approximation of the point distances, i.e. also the density structure of the embedded trajectories.

**Comment 8**

Line 193. The intuitive meaning of the 'generating distances' that are not being used here should be mentioned

**Answer to comment 8**

Than you for the comment. In the new version, we briefly mention what a finite generating distance would mean.

For $\delta \in \mathbb{R}$, the $\delta$-neighbourhood of a point $p \in \mathbb{R}^M$ is defined as the $M$-dimensional ball of radius $\delta$ around $p$. Define $M_\varepsilon(p)$
* * *
$M_\delta(p)$ as the number of points that is in the $\varepsilon\delta$-neighbourhood of $p$, including $p$ itself. OPTICS requires one parameter, an integer $s_{min}$ (called MinPts by Ester et al. (1996)Ankerst et al. (1999)), that defines the *core-distance* of a point $p$ as

$$\quad c(p) = \{\min(\varepsilon\delta) \mid M_{\varepsilon\delta}(p) \geq s_{min}\}. \tag{6}$$

The core distance is simply the minimum radius of a ball around $p$, such that the ball contains $s_{min}$ points. Note that the generating distance that we set to infinity is a maximum cut off distance for the computation of the core distance in eq. (6), beyond which the core distance is not defined. As we do not have an intuition for a good value of such a cut off, we remove it by setting it to infinity.
* * *
**Comment 9**

Line 196. The definition of the epsilon neighborhood appears incomplete. Is it not the M-dimensional sphere of radius epsilon? Otherwise, what is the epsilon?

**Answer to comment 9**
Indeed the epsilon-neighborhood of p is just the M-dimensional ball around the point p, and the previous version was incomplete. We have changed this in the new version, together with renaming epsilon to delta, see our answer to comment 8.
* * *
**Comment 10**
Line 200. It would be very helpful to write out in words the meaning of Eq. (6). My understanding is that c(p) is minimum distance epsilon such that the number of points in an epsilon neighborhood is greater than a specified number.

**Answer to comment 10**
Thank you for your comment. Your interpretation was correct. We have made it more clear in the new version, see the answer to comment 8.
* * *
**Comment 11**
Line 213. I did not immediately understand how it arises that there are valleys in the reachability if you have sorted iteratively on the reachability. You might explain that this happens as you encounter

groups of points that are all near to each other, thus replacing earlier high values of reachability with lower values.

**Answer to comment 11**

Thank you for the comment. Indeed, it is the sorting that is the most important step in the algorithm. We added some more explanation in the new version.

> Note that the ordering of points is achieved by constantly updating the ordered seed list, cf. step 3. In this way, the algorithm
> 275 iterates through groups of dense points one after the other, and only continues with other points once a dense region has been fully explored. Note also that the entire algorithm depends on the choice of the parameter $s_{min}$. The value of $s_{min}$ should be chosen roughly as a minimum value of the expected cluster size. In the examples presented in this paper, we take values for $s_{min}$ that correspond to the estimated minimum size of the coherent sets.

**Comment 12**

Line 216. The phrasing here made me wonder if this was a second, different epsilon. It would be clearer to say that you choose a value for the parameter epsilon. Also, it appears this is conditional on a choice of s_min which should then be emphasized.

**Answer to comment 12**

Thank you for very much for pointing this out. Indeed, this was a second epsilon, and the presentation in the first version was confusing. We have made the appropriate changes in the new version by re-naming one of the epsilons into delta. See our answer to comment 8.

**Comment 13**

Line 228. What are the permissible values of k in condition (a)?

**Answer to comment 13**

We have made this more precise in the new version. It can be any integer larger than zero and smaller than N - l.

**Comment 14**

Figure 2, what are the units of the y-axis in the left column of plots?

**Answer to comment 14**

Thank you for this comment. Indeed, we missed to specify the units of all reachability values. We do so in all figures in the new version (apart from the network embedding case in the appendix, where quantities are dimensionless), see the example below.

[Figure]
* * *
**Comment 15**

Figs 2 and 3, some of the colored dots lie above the epsilon threshold.

**Answer to comment 15**

This is correct, DBSCAN classifies the points below the line only up to boundary points, i.e. there can be points at the cluster boundary that belong to the cluster. We have made this more clear in the new version.

**4 Results**

**4.1 Bickley jet flow**

375 We start with the direct embedding of the  Bickley jet flow trajectories, cf. section 2. The data matrix has dimension  $X \in \mathbb{R}^{12,000 \times 123}$. We apply the OPTICS algorithm to the resulting points  together with DBSCAN clustering, choosing $s_{min} = 80$ as a minimum size of the finite-time coherent sets. In the following, all axis units are in multiples of 1000 km. Figure 2 shows the reachability plot, together with the DBSCAN clustering result of three different choices of $\epsilon$. The six vortices and the jet are clearly visible as the major valleys in the reachability plot. The hierachical
* * *
380 structure of the DBSCAN clustering with decreasing $\epsilon$ is visible in the figures from top ( large-scale coherence) to bottom ( small-scale coherence).  Note that for the DBSCAN clustering  results, boundary points of the clusters can be above the hozitonal line at $y = \epsilon$. This is because of the definition of the DBSCAN clustering in section 3.3.

**Comment 16**

Figure 4. I really don't understand the two dimensions of these plots, nor the star-shaped patterns, could you explain these more?

**Answer to comment 16**

We have now made the presentation of the methods regarding classical MDS more clear, also relating it to principal component analysis, see the answer to your comment 7. In addition, we have provided more explanation on the star-shaped structure in the results section.

> 385 To illustrate the difference between OPTICS and k-Means, we use the embedded trajectories and apply classical MDS to obtain a 2-dimensional embedding. As  described in section 3.2.2, this assures to capture the major variance along the embedding axes. The spectrum of $B$ in eq. (4) is shown in fig. A1 in the appendix, with two clearly dominant eigenvalues.
> 390 The fact that there are two very dominant eigenvalues assures that the illustration of the data in the plane captures the major variance of the data points. Figure 3a shows the corresponding embedding of the trajectories in the 2-dimensional  Euclidean space. The star-shaped distribution of data points reflect the strong

> symmetries of the underlying idealized Bickley jet flow. Such symmetry is not expected to be present for more realistic flows. Figures 3b and 3c show the cluster labels for OPTICS with DBSCAN clustering at
> 395  $\epsilon = 10^6$ km, and for a k-Means clustering with $K = 8$ clusters,  respectively. $K = 8$ corresponds to the six vortices, the jet, and one noise cluster as suggested by Hadjighasem et al. (2016).

In addition, we have further discussed the failure of k-Means in relation to the star-shaped structure of the embedding.

> The corresponding clustering  results in real space are shown in figs. 4 and 5
> for OPTICS and k-Means, respectively. The jet and the six vortices are clearly recognizable as dense accumulations of points in
> 400 the 2-dimensional space of fig. 3b, see fig. 4 for the corresponding colours. The clustering result with k-Means in fig. 5 shows that the clusters corresponding to the vortices are much less focussed. In addition, each of the eight clusters in fig. 3c contains some of the noisy points of fig. 3b, which shows that using one additional cluster for noise does not   work in this situation. It is interesting to note that capturing the noisy data points of fig. 3b by an additional cluster in k-Means is geometrically impossible, simply because k-Means clusters
> 405 are circular. Covering all noisy points without including the centre, i.e. the jet in fig. 3b, is not possible for k-Means.

**Comment 17**

Data locations at Zenodo should be cited, not only the papers referring to them.

**Answer to comment 17**

The reference is actually a Zenodo link, not a paper. Note that there were two references Wichmann 2020 (Zenodo link) and Wichmann et al. (2020) (previous paper). In the new version, there is now also a Zenodo link to an animation for the Agulhas flow.

**Comment 18**

Throughout the paper, the authors consistently omit the subject ahead of an infinitive, e.g. "which allows to detect". I believe this is grammatically incorrect (in US usage anyway). "allows one to detect" or "allowing the detection of" sound better

**Answer to comment 18**

Thank you, we have made appropriate changes in the new version.
* * *
**Comment 19**

l 42 and 90. "sparse" should probably be used instead of "scarce". The former means thinly distributed while the latter means hard to come by.

**Answer to comment 19**

We have made appropriate changes in the new version.
* * *
**Comment 20**

l 128. NumPy and Zenodo are the standard capitalizations

**Answer to comment 20**

We made the suggested changes in the  new version. Thank you for noting.
* * *
**Comment 21**

l 141. "method" should be "methods"

**Answer to comment 21**

Thank you for noting, we corrected it in the new version.
* * *
**Comment 22**

l 156. Straightforward

**Answer to comment 22**

Thank you for noting, we corrected it in the new version.
* * *
**Comment 23**

l 191. "and as will become clear"

**Answer to comment 23**

Thank you for noting, we corrected it in the new version.
* * *
**Comment 24**

l 217. "is equal to" should be "set of points is equivalent to".

**Answer to comment 24**

Thank you for noting, we corrected it in the new version.
* * *
**Comment 25**

l 243. "a priory" should be "a priori"

**Answer to comment 25**

We corrected it in the new version.
* * *
**Comment 26**

l 279. "large- and small-scale"

**Answer to comment 26**

Thank you for noting, we corrected it in the new version.
* * *
**Comment 27**

l 354. GitHub

**Answer to comment 27**

Thank you for noting, we corrected it in the new version.
* * *
**Comment 28**

l 359. There is a title of an appendix with no appendix.

**Answer to comment 28**

The content of appendix C consisted of only two figures, C1 and C2. It appeared as without content due to the page break. In the new version, we have removed one appendix as we include the figures in the main text, such that the formatting looks better.
* * *
**Comment 29**

l 360 & 361. "particle-based"

**Answer to comment 29**

Thank you for noting, we corrected it in the new version.
* * *
**Comment 30**

l 383. "ot"

**Answer to comment 30**

Thanks for the careful read, we made the changes in the revised manuscript.
* * *
**Comment 31**

l 389. There should be a period at the end of this sentence

**Answer to comment 31**

Done. Thanks for noting.
* * *
**Comment 32**

Figure C1, "three" eigenvalues should be "two", correct?

**Answer to comment 32**

Yes, indeed. Thanks for reading also the appendix figure captions so carefully! We corrected this in the new version.

---

## Author Comment (AC2)

**Answer to reviewer 2**

**General answer:**
We thank the reviewer for the critical comments, and in particular for the detailed analysis of other methods and their comparison to our approach. We agree with most points raised by the reviewer. We have made major adaptations to the formulations in the revised version, and explain the relation of our method to existing studies in more detail.

**Please note:**
The images in this file are excerpts of the revised version in latexdiff. Please apologize the formatting problems of latexdiff that cuts off references at line breaks. This is not the case in the revised version.
* * *
**Comment 1**
There are already several clustering methods in the literature for finding finite-time coherent sets, including a density-based clustering DBSCAN by Schneide-etal'18, which is a special case of the OPTICS approach in the manuscript. The idea of a hierarchy of finite-time coherent sets has been considered by Ma/Bollt'13. The paper Fr/Sa/Ro'19 develops a robust method to classify only those sets are that coherent, not fully partitioning the domain. In Fr/Sa/Ro'19, coherent sets at different spatial scales are also considered, similar to a hierarchy. Fr/Sa/Ro'19 also considers the Bickley jet and ocean eddies, with ocean eddies listed as a motivation in Fr/Sa/Ro'19 for developing a non-partitioning approach. Not limited to the work above, I would say there is some "upselling" of the novelty in the manuscript, and that prior work is occasionally omitted, mischaracterized, or overly criticized.

**Answer to comment 1**
Thank you for this comment. We did not intend to upsell our work, or omit, mischaracterize or overly criticize existing work. In fact, our work has been majorly motivated by the paper of Froyland et al. 2019. But we understand that the original manuscript appeared to do so, and we thank the reviewer to making this clear to us. We have made the following changes in the new version.

1. We mainly removed the discussion of other methods in the introduction and moved it to a separate section. In the introduction, we emphasize that our work is majorly inspired by Froyland et al. 2019. We are also more specific about the actual problem at hand, i.e. the detection of many small scale coherent sets in large-scale, noisy ocean flows.

[revised manuscript text omitted]

A positive aspect is that the (standard) "DBSCAN" and "\xi" clustering outputs of the OPTICS clustering could provide potentially useful hierarchical information, and to my knowledge this is a new way of analyzing the dynamics. Unfortunately, this is not explored much, and the authors do not provide an intuitive explanation of what the "DBSCAN" and "\xi" clustering algorithms are actually doing in their dynamical context. It would be beneficial for the authors to link the algorithms more with the dynamical inputs (trajectories) and the dynamical problem being solved. As this is the main contribution of the paper, I think this needs to be expanded much more. The reasons behind the choices of which clustering algorithm is applied to the different datasets should also be explained.

**Answer to comment 2**

Thank you for this comment. We were indeed lacking some form of intuition behind the two clustering methods and their application. We have made the following changes.

1.  More explanation about the embedding and why the embedded trajectories create a signal in terms of data density.

**3.2 Trajectory embedding**

**3.2.1 Direct embedding**

The direct embedding of each trajectory in $\mathbb{R}^{DT}$ is the most  straightforward embedding as it requires no further pre-processing of the trajectory data. For simplicity, assume we are given a set of $N$ trajectories in a 3-dimensional space, i.e.

200   $(x_i(t), y_i(t), z_i(t))$ where $i = 1, \ldots, N$ and $t = t_1, \ldots, t_T$. We then simply define the embedding of trajectory $i$ in the abstract $3T$-dimensional space as

$$u_i = (x_i(t_0), x_i(t_1), \ldots, x_i(t_T), y_i(t_0), y_i(t_1), \ldots, y_i(t_T), z_i(t_0), z_i(t_1), \ldots, z_i(t_T)) \in \mathbb{R}^{3T}, \tag{3}$$

and impose an Euclidean metric in $\mathbb{R}^{3T}$ to measure distances between different embedded trajectories. The resulting embedded data matrix $\bar{X}$ is then simply given by the vertical concatenation of the different embedding vectors. This kind of

205   embedding was also explored by Froyland and Padberg-Gehle (2015), together with a fuzzy-c-means clustering. Intuitively, if two trajectories $i$ and $j$ belong to the same finite-time coherent set, the corresponding particles follow very similar pathways, i.e. the Euclidean distance of the embedding vectors $d_{ij} = \|u_i - u_j\|$ is expected to be small. On the other hand, a particle $i$ that belongs to a coherent set is expected to have a larger distance to a particle $j$ that is not part of the set. In other words, groups of particles that form a finite-time coherent set are *dense* in the embedding space. This motivates to use a density-based

210   clustering algorithm to detect finite-time coherent sets.

To take into account the $\pi r_0$-periodicity in x-direction of the Bickley jet flow, we first put the individual 2-dimensional data points on the surface of a cylinder with radius $r_0/2$ in $\mathbb{R}^3$, and interpret the resulting  trajectories in a 3-dimensional Euclidean space. The resulting data matrix is $\bar{X} \in \mathbb{R}^{N \times 3T}$, with $N = 12,000$ and $T = 41$. For the Agulhas particles, we put the single data points on the earth surface in a 3-dimensional Euclidean embedding space by the standard coordinate transformation

215   of spherical to Euclidean coordinates. The resulting data matrix is thus $\bar{X} \in \mathbb{R}^{N \times 3T}$ with $N = 23,821$ and $T = 21$.
* * *
2.   An intuitive explanation of the two clustering methods and their major properties.

Intuitively, the two clustering methods can be understood as follows. DBSCAN detects those groups of points that have a certain minimum density defined by the minimum reachability distance $\epsilon$. Clusters detected by DBSCAN are therefore defined by a global density criterion. This assumes no structural differences in the type of coherent sets in different regions of the fluid. Different from that, the $\xi$-clustering method detects clusters by finding strong changes in the density of the data points, and not

320   based on absolute densities. This has the advantage that clusters of different absolute density can be detected. Such a situation can arise if the distribution of particles is inhomogeneous over the fluid domain, or if the spatial extend of the fluid domain is very large such that the properties of finite-time coherent sets vary significantly. It is important to note that the main result of OPTICS is the reachability plot itself. The DBSCAN- and $\xi$-clustering methods should be seen as useful tools to identify the most important features of that plot.
* * *
3. We have included a DBSCAN clustering result in the main figure of the Agulhas flow example, and discuss the differences between xi and DBSCAN clustering.

[Figure]

**Figure 6.** Result of the OPTICS algorithm applied to the direct embedding of the trajectories, with different clustering methods. Grey particles correspond to noise.

435  Figure 6 shows that for this situation, the $\xi$-clustering method detects more Agulhas rings than DBSCAN. While the clustering results shown in the figure all depends on the parameter values for $\xi$ and $\epsilon$, it is visible in the reachability plot of fig. 6g that the definition of some eddies includes the entire boundary of the valleys, i.e. up to very high reachability values. At the same time, the detection of the large-scale clusters as in 6a-c is not possible with the $\xi$-clustering method. These findings are in fact expected, cf. the discussion of the two clustering methods at the end of section 3.3. DBSCAN is best to detect global

440 density structures, i.e. when the reachability values of all points are compared to the same cut-off $\epsilon$. Regions that are dense locally but not necessarily globally are better detected with the $\xi$-clustering method. Despite these differences between the two clustering methods, we again emphasize that the main result of OPTICS is the reachability plot itself. Fig. 7 shows a colour

17

map at initial time of the reachability values. We clearly see Agulhas rings as the dark regions corresponding to lowest values of reachability. The regions of large reachability correspond to trajectories that are relatively noisy compared to all the other

445 trajectories.
* * *
**Comment 3**

The (uncited) paper Froyland/Junge'18 develops a finite-element approximation of the dynamic Laplacian, which is a very accurate and robust method of finite-time coherent set extraction for low-dimensional systems of the type treated in the Wichmann manuscript. In Froyland/Junge'18 there are no free parameters, the method is unaffected by the density of the data points, and estimates are produced on the whole domain. A comparison can be made for the Bickley example in the Wichmann manuscript because the setup is identical. Wichmann et al uses a 200x60 grid of points and particle positions at times t=0, 1, 2, 3,..., 39, 40. Froyland/Junge'18 studied the same Bickley flow as in Wichmann, except that Froyland/Junge'18 used a coarser 100x30 grid of points and only particle positions at time 0 and time 40. Figure 15 in Froyland/Junge'18 shows much clearer images with fewer trajectory inputs. Thus, I think there is not a strong case for the approach in the manuscript being a better performer.

**Answer to comment 3**

Thank you for this comment, and we apologize for not having cited that paper. Note however that the clustering results presented there are also based on k-Means clustering, and there are no free parameters only up to the choice of embedding dimension and the number of clusters. The paper also shows that the approach with k-Means works for situations where the coherent sets are not very small compared to the fluid domain, see the problems of k-Means in this context in the paper by Froyland et al. 2019. Nevertheless, the concepts presented there are powerful, as they provide a type of embedding that has a clear dynamical motivation, which is an advantage compared to our heuristic embedding. We refer to the paper at many places in the new version in different contexts:

1. End of the new section on comparison to other methods

   A downside of our method compared to other approaches is the rather ad-hoc choice of embedding, cf. eq. (3). Different from
   370  many other methods, most notably the ones of Banisch and Koltai (2017), Froyland and Junge (2018) and Froyland et al. (2019)
   , this type of embedding is not derived from a meaningful dynamical operator. It could be fruitful to explore a combination of
   these more meaningful embeddings together with OPTICS as a clustering algorithm in future research.

2. We now also tested our method with the Bickley jet using less particles and less data points for each trajectory. Our method does indeed not perform as well as the method of Froyland and Junge (2018), and we want to thank the reviewer for explicitly mentioning this possible comparison.

   We finally also tested the performance of our algorithm with a random subset of 2,000 particles, using data for every five
   days instead of every day, cf. fig. A1 in the appendix. OPTICS still detects the six vortices and the jet, although the cluster
   boundaries are less clearly defined compared to fig. 2. Froyland and Junge (2018) detect the vortices and the jet by using data

   **15**

   of 3,000 particles only at initial and final times ($t = 0$ and $t = 40$ days). Our method is not able to detect the expected finite-time
   415  coherent sets with using only initial and final particle data. This is likely to be a result of the ad-hoc direct embedding, cf. eq.
   (3), see the discussion at the end of section 3.4.

[Figure]

Figure A1. Result of  the OPTICS algorithm for a random subset of 2,000 particles in the  Bickley jet flow,  with particle data every 5 days instead of every day.  To account for the smaller number of particles, we set $s_{min} = 15$ for this case.  The six vortices and the jet  still clearly visible.

3. In the conclusion, we come back to the problems of our form of embedding and mention again that a combination of the embedding of Froyland and Junge (2018) together with OPTICS could yield better results.

490  We apply OPTICS to Lagrangian particle trajectories directly, in the spirit of Froyland and Padberg-Gehle (2015). OPTICS successfully detects the expected coherent structures in the Bickley jet model flow, separating the six vortices and the jet from background noise. We also apply  OPTICS to simulated trajectories in the eastern South Atlantic and successfully identify Agulhas rings, separated by noise. We visualize the difference  between OPTICS and k-Means with a 2-dimensional embedding of the trajectories based on classical multidimensional scaling. We also show how

495  OPTICS can be applied to the spectral embedding of the  particle-based network proposed by Padberg-Gehle and Schneide (2017), providing a necessary amendment to  their method to detect coherent vortices in a large ocean domain, i.e. when k-Means fails. Our method is

500

**20**

 very simple to implement in Python, as OPTICS is available in the SciPy sklearn package. While we here present the results of OPTICS with three different kinds of  embeddings, it is likely that OPTICS also works for other trajectory embeddings,  such as the spectral embeddings

505  of Banisch and Koltai (2017) or Froyland and Junge (2018). Using such dynamically motivated embeddings instead of the ad-hoc direct embedding presented here could be a promising direction for future research.
* * *
**Comment 4**

The idea to not fully partition the domain has already been treated in Fr/Sa/Ro'19. Regarding the ocean eddy example in the manuscript, Fr/Sa/Ro'19 also applied the method of Froyland/Junge'18 to ocean flow and successfully extracted a greater number of eddies than Wichmann at a higher quality. On the other hand, Fr/Sa/Ro'19 used AVISO-derived trajectories rather than model output, so it could be that Wichmann is using a rougher velocity field. Wichmann also used lower trajectory density than Fr/Sa/Ro'19 by a factor of about 4; both of these items could make Wichmann's task more difficult, compared to Fr/Sa/Ro'19.

**Answer to comment 4**

Thank you for pointing this out. For a detailed comparison of the both methods, it would indeed be necessary to choose exactly the same flows. Detecting a greater number of eddies in a specific ocean domain does not necessarily have an implication for the usefulness of a method. We would like to note again that the results of Froyland et al. (2019) were a major motivation for our paper, and we do not aim to compete with their method any aspects. We would rather like to show how a change of clustering algorithm, instead of a change of embedding, can also yield better results compared to partition-based clustering, see the paragraph below in the revised paper on the comparison to other methods. We believe that a combination of the embedding of Froyland and Junge 2018 together with OPTICS could be a useful extension of our method. See our answer to your comments 1 and 3 for more content relating to their method.